# Systematic Review of Molecular Targeted Therapies for Adult-Type Diffuse Glioma: An Analysis of Clinical and Laboratory Studies

**DOI:** 10.3390/ijms241310456

**Published:** 2023-06-21

**Authors:** Logan Muzyka, Nicolas K. Goff, Nikita Choudhary, Michael T. Koltz

**Affiliations:** Department of Neurosurgery, Dell Medical School, The University of Texas at Austin, 1501 Red River Street, Austin, TX 78712, USA

**Keywords:** glioma, glioblastoma, molecular targeted therapy, WHO brain tumor guideline, idh-mutant glioma, astrocytoma, idh-mutant astrocytoma, oligodendroglioma, protein kinase pathway, microenvironmental targets, immunotherapy, cell cycle, apoptosis, wnt/β-catenin pathway, molecular pathway, clinical studies, animal studies, systematic review, brain tumor

## Abstract

Gliomas are the most common brain tumor in adults, and molecularly targeted therapies to treat gliomas are becoming a frequent topic of investigation. The current state of molecular targeted therapy research for adult-type diffuse gliomas has yet to be characterized, particularly following the 2021 WHO guideline changes for classifying gliomas using molecular subtypes. This systematic review sought to characterize the current state of molecular target therapy research for adult-type diffuse glioma to better inform scientific progress and guide next steps in this field of study. A systematic review was conducted in accordance with PRISMA guidelines. Studies meeting inclusion criteria were queried for study design, subject (patients, human cell lines, mice, etc.), type of tumor studied, molecular target, respective molecular pathway, and details pertaining to the molecular targeted therapy—namely the modality, dose, and duration of treatment. A total of 350 studies met the inclusion criteria. A total of 52 of these were clinical studies, 190 were laboratory studies investigating existing molecular therapies, and 108 were laboratory studies investigating new molecular targets. Further, a total of 119 ongoing clinical trials are also underway, per a detailed query on clinicaltrials.gov. GBM was the predominant tumor studied in both ongoing and published clinical studies as well as in laboratory analyses. A few studies mentioned IDH-mutant astrocytomas or oligodendrogliomas. The most common molecular targets in published clinical studies and clinical trials were protein kinase pathways, followed by microenvironmental targets, immunotherapy, and cell cycle/apoptosis pathways. The most common molecular targets in laboratory studies were also protein kinase pathways; however, cell cycle/apoptosis pathways were the next most frequent target, followed by microenvironmental targets, then immunotherapy pathways, with the wnt/β-catenin pathway arising in the cohort of novel targets. In this systematic review, we examined the current evidence on molecular targeted therapy for adult-type diffuse glioma and discussed its implications for clinical practice and future research. Ultimately, published research falls broadly into three categories—clinical studies, laboratory testing of existing therapies, and laboratory identification of novel targets—and heavily centers on GBM rather than IDH-mutant astrocytoma or oligodendroglioma. Ongoing clinical trials are numerous in this area of research as well and follow a similar pattern in tumor type and targeted pathways as published clinical studies. The most common molecular targets in all study types were protein kinase pathways. Microenvironmental targets were more numerous in clinical studies, whereas cell cycle/apoptosis were more numerous in laboratory studies. Immunotherapy pathways are on the rise in all study types, and the wnt/β-catenin pathway is increasingly identified as a novel target.

## 1. Introduction

As the most common brain tumor in adults, gliomas have sustained the focus of scientific research for the past several decades. Recently, more attention has been drawn to the diagnostic criteria of gliomas with the restructured 2021 WHO Classification of Tumors of the Central Nervous System, specifically focusing more on molecular biomarkers as a means of categorization [1]. Within this classification adult-type diffuse gliomas are the most prevalent tumor types, defined on the basis of molecular expression of isocitrate dehydrogenase (IDH) and the 1p/19q codeletion. These glioma subtypes include astrocytoma (IDH-mutant astrocytoma), oligodendroglioma (IDH-mutant and 1p19q-codeleted), and glioblastoma (GBM) (IDH-wildtype) [1]. The typical management of adult-type diffuse glioma begins with a resection or biopsy, followed by possible radiotherapy and/or chemotherapy with the alkylating agent, temozolomide, or the combination procarbazine, lomustine, and vincristine (PCV) [2]. Even with this regimen, recurrence is prevalent, and the prognosis is dismal, particularly in GBM, which has an average survival of 14–16 months [3].

As gliomas are becoming more molecularly defined, so too is their treatment progressing more towards the targeting of molecular pathways [4]. Compared with traditional chemotherapeutic drugs, molecularly targeted antitumor therapy has the advantage of strong specificity with minimal damage to normal tissues. Molecular-targeted glioma therapies have gained traction in the scientific literature, with many analyses centered on identifying mechanisms pertinent to glioma growth [5]. The Raf/MEK/Erk pathway has been of particular interest as a targetable pathway due to its preponderance among gliomas [5]. Additionally, a systematic review by Da Silva et al. highlighted the molecular targeted therapies in clinical trials for GBM, identifying four categories of targets: targeting the potential for unlimited replication, growth autonomy and migration, cell cycle and apoptosis, and angiogenesis [6].

To date, there has yet to be a systematic review of the literature characterizing molecular targeted therapy in adult-type diffuse gliomas. A comprehensive understanding of the progress in this field—both in terms of existing therapies and novel targets—is integral to guiding advancement in treatment development, integration into clinical trials, and more adequate treatment options for diffuse glioma patients.

## 2. Methods

A systematic review was performed, characterizing the current state of molecular targeted therapies for gliomas, to inform scientific progress and guide advancement in this area of study. The protocol was conducted in accordance with PRISMA (preferred reporting items for systematic reviews and meta-analyses) guidelines.

### 2.1. Search Strategy

A literature search of English-text articles was conducted through January 2023 using PubMed and Web of Science. Categories of concepts related to both molecular targeted therapy and glioma, both adhering to the 2021 WHO classification as well as prior classifications (including language such as low- or high-grade glioma), were searched; results were combined via Boolean operators (Appendix B).

Additionally, a search with the same search terms was conducted on clinicaltrials.gov to assess clinical trials relating to molecular targeted therapy for adult-type diffuse glioma.

### 2.2. Selection Criteria

Article titles and abstracts were screened for relevance by two authors (L.M. and N.K.G.), and duplicates were removed. The remaining articles were then screened in full text by three authors (L.M., N.K.G., and N.C.). Inclusion criteria used were: any clinical or laboratory studies testing molecular targeted therapies for glioma or laboratory studies identifying novel molecular targets for glioma with a target of adult-type diffuse glioma or its subtypes. Exclusion criteria used were: brain tumors other than adult-type diffuse glioma—such as medulloblastoma; any study of pediatric tumor focus, etc.; papers centering on the delivery technology rather than the molecular target; systemic therapies or adjuvants to molecular targeted therapies; studies with no molecular target identified; or papers that were correspondences, reviews, or commentaries. Conflict resolution at all stages of article selection was via discussion between authors.

### 2.3. Data Extraction

The data extraction for the systematic review included the following: first author, year of publication, design (multi-institutional retrospective analysis, in vivo, in vitro, and in vivo, etc.), study subject (patients, human cell lines, mice, etc.), ages of subjects, type of tumor studied, molecular target, respective molecular pathway, and the modality and results of the molecular targeted therapy investigated.

The data extraction for clinicaltrials.gov was limited to ongoing trials—defined as those with a status of completed, recruiting, or active, non-recruiting. The extracted variables included the following: title of the study, year started and year of most recent update, tumor type, NCT number, sponsoring or collaborating organization, molecular target of interest, intervention utilized, as well as the phase of the study (Phase 1, 2, or 3), status (active or recruiting), funding sources (NIH, industry, or other), and results, if available.

### 2.4. Data Categorization

Published studies were then divided into three main categories: clinical studies testing existing molecular targeted therapies; laboratory studies testing existing molecular targeted therapies; and laboratory studies identifying a novel molecular target.

Unless specifically stated otherwise in the study, tumor types were classified by molecular associations with respective cell lines in the literature. For instance, those classified under GBM included cell lines known to harbor wild-type IDH (U87, U251, T98G, and A172) and human tumors classified specifically as GBM [7,8,9,10,11]. In instances where molecular mutation information was not readily available or numeric glioma grading was utilized (grades I–IV), these were classified as simply “Glioma”.

To assess for 3-dimensional (3D) or spheroid technologies in laboratory studies of existing therapies, a full text search of terms related to these technologies—“sphere”, “spheroid”, “3D”, “3-D”, “3-Dimensional”—was also conducted.

Ongoing clinical trials were also queried in this manner and organized by tumor type.

### 2.5. Statistical Analysis

A meta-analysis was not conducted; therefore, descriptive data is reported for most variables in this study. To compare means by group, ANOVA testing was utilized. The chosen type 1 error rate was set to *p* < 0.05. All statistical analyses were performed via IBM SPSS Statistics for Macintosh (version 28.0.1.1) (Armonk, NY, USA).

### 2.6. Quality Assessment

The quality of evidence was determined by study design and graded using a level of evidence scheme adapted from Ackley et al. (Table 1) [12].

## 3. Results

### 3.1. Search Results

This study identified 350 articles for inclusion. (Figure 1) Data extraction for each respective category is detailed. Only 15% (52/350) of the total articles were clinical studies (Table 2). The majority of articles (54%, 190/350) were laboratory studies investigating existing molecularly targeted therapies (Table 3), and 31% (108/350) were laboratory studies identifying new molecular targets without testing an existing therapy (Table 4). Across these groups, clinical studies had a more recent median publication year (2017) compared to both laboratory studies testing existing therapies (2015) and laboratory studies identifying novel targets (2016; *p* < 0.05).

### 3.2. Clinical Studies Implementing Molecular Targeted Therapies

Fifty-two clinical studies implementing molecular targeted therapies for glioma were identified, with a median publication year of 2017 (Table 2) [343]. In terms of tumor type, 40/52 (77%) studied GBM (GBM), 10/52 (19%) studied IDH-mutant astrocytoma, and there was one study on an IDH-wt, 1p19q co-deleted glioma (2%). In terms of molecular targets, 26/52 (51%) targeted some form of protein kinase, 15/52 (29%) targeted angiogenesis or environmental pathways, 3/52 (6%) targeted immunotherapy pathways, and 3/52 (6%) targeted cell cycle or apoptosis pathways.

The level of evidence for the clinical studies varied. Eight studies had Level II evidence (15%), as they were multi-institutional clinical trials. Most published clinical studies had Level IV evidence (34/52; 65%), consisting of single institutional phase II or prospective trials. The rest of the clinical studies (10/52, 19%) were case reports or series, classifying them as Level VI studies (Table 1 and Appendix A).

### 3.3. Laboratory Studies Implementing Molecular Targeted Therapies

There were 190 laboratory studies implementing existing molecularly targeted therapies for glioma (Table 3). An overwhelming majority of studies (167/190, 88%) focused on GBM, followed by studies of unspecified gliomas (17/190, 9%), then IDH-mutant astrocytomas (11/190, 6%), then oligodendrogliomas (3/190, 2%), with some studies covering multiple glioma types. The most prevalent molecular targets were those involving protein kinase pathways (79/190, 42%), particularly tyrosine kinase receptors. Out of the protein kinase pathways, PI3K/Akt/mTOR, Ras/BRAF/Mek/Erk, and upstream targets were found in the largest proportion (29/79, 37%). Additionally, nearly a quarter of all clinical studies targeted cycle/apoptosis/transcription-targeted pathways (41/190, 22%). The next most prevalent pathway targets were microenvironmental targets (30/190, 16%)—including angiogenesis, cell-cell adhesion molecules, and iron/cation regulation—followed by immunotherapy pathways (8/190, 4%) (Table 3). All studies were Level III evidence (Table 1).

Most studies (100/190, 53%) were conducted using combined in vitro and in vivo designs; the next most common were 61/190 (32%) in vitro studies, then 6/190 (3%) were combined in vivo and ex vivo studies (Appendix A). The most frequent cell lines were U87 (105/190, 55%), U251 (51/190, 27%), T98 (22/190, 12%), A172 (19/190, 10%), or GBM patient samples (28/190, 15%) (Appendix A).

Of the laboratory studies testing existing molecular targeted therapies, all were queried for whether or not they utilized spheroid or 3-dimensional (3D) technologies for cell culture as part of their methodology. Fifty-nine (31%) of studies adopted tumor sphere or 3D technology (Appendix A).

### 3.4. Laboratory Studies Identifying Novel Molecular Targets

There were 108 laboratory studies identifying novel molecular targets for treating glioma (Table 4).

The majority of studies (82/108, 76%) focused on GBM, followed by 29/108 (27%) studying unspecified gliomas, 6/108 (6%) studying IDH-mutant astrocytoma, and lastly 3/108 (3%) studying oligodendroglioma, with some studies covering multiple glioma types. Twenty-seven (25%) studies targeted protein kinase pathways, 21/108 (19%) targeted cell cycle/apoptosis pathways, 16/108 (15%) studies targeted microenvironmental targets, 10/108 (9%) studies targeted immunotherapy pathways, and 7/108 (6%) targeted the wnt/beta catenin pathway (Figure 2). All studies were Level III evidence (Table 1).

### 3.5. Ongoing Clinical Trials

A search of clincialtrials.gov yielded 341 clinical trials, of which 119 met our inclusion criteria for ongoing clinical trials investigating molecular targeted therapies for adult-type diffuse glioma and its subtypes (Table 5). The most prevalent targets involved protein kinase pathways (65/119, 55%), followed by angiogenesis or microenvironmental targets (33/119, 28%), then cell cycle/apoptosis (10/119, 8%), and immunotherapy pathways (10/119, 8%). For tumor types, 74/119 (62%) tested GBM, 5/119 (4%) tested IDH-mutant astrocytoma, and 2/119 (2%) tested oligodendroglioma, with many studies testing specific subcategories. The average start year was earlier in trials testing protein kinase targets (2009 ± 6), compared with trials testing cell cycle/apoptosis inhibitors (2016 ± 4) and immunotherapies (2018 ± 2), which occurred more recently on average (*p* < 0.001).

The most common funding source was industry-related funding (54/119, 45%), followed by the National Institute of Health (NIH) (45/119, 38%) (Appendix A). All ongoing clinical trials were in phase I or II.

## 4. Discussion

This systematic review examined the current evidence on molecular targeted therapy for adult-type diffuse glioma. The majority of clinical and laboratory studies focused on GBM, with few studies examining IDH-mutant astrocytomas, oligodendrogliomas, or unspecified gliomas. In both clinical and laboratory settings, protein kinase pathways—particularly PI3K/Akt/mTOR and Ras/BRAF/Mek/Erk—were the most commonly targeted molecular pathways. The next most common molecular targets in published clinical studies and clinical trials were microenvironmental targets—including angiogenesis, cell-cell adhesion, or ion/cation regulation—followed by cell cycle/apoptosis pathways and immunotherapy. The second most common molecular targets in laboratory studies were cell cycle/apoptosis pathways, followed by microenvironmental targets, and then immunotherapy pathways. The wnt/β-catenin pathway was also prevalent in the studies identifying novel targets. The level of evidence for published clinical studies varied, with the majority being Level IV—consistent with early-phase, single-institution clinical trials; all laboratory studies were quasi-experimental designs. Published clinical studies testing molecular targeted therapies, in general, were published more recently than laboratory studies. Lastly, clinical trials on protein kinase pathways began earlier than other clinical trial types, particularly trials testing cell cycle/apoptosis targets or immunotherapy.

### 4.1. Adult-Type Diffuse Glioma Subtypes

Though the overwhelming majority of studies centered on GBM, the literature shows that adult gliomas found more frequently in practice tend to harbor IDH mutations [7,344]. The reason for the overrepresentation of GBM-focused studies and the underrepresentation of IDH-mutant astrocytoma or oligodendroglioma is multifactorial. First off, the updated WHO classification is a recent development as of 2021; because the majority of the works in this study occurred prior to the molecular subtype differentiation, there were likely studies that self-identified as GBM studies that may have included tumors with an IDH mutation or 1p19q co-deletion. To the best of our ability, we retroactively identified studies that specifically identified these molecular statuses, but those were few in number. Additionally, it is likely that GBM has received more research funding and scientific attention than other brain tumors, perhaps due to its more aggressive nature and mortality rates. Therefore, the funding for studies investigating IDH-mutant astrocytoma or oligodendroglioma may be less robust. Of note, the ongoing clinical trials for glioma vastly favor GBM as well, receiving the majority of funding from industry sources. Further studies to quantify the distribution of research funding between glioma subsets would be necessary to confirm this association. Lastly, the standard cell lines for all glioma research tend to be glioblastoma models, particularly U87, U373, and U251, as also reflected in our study [345] (Appendix A).

### 4.2. Protein Kinase Pathways

In terms of molecular targets, protein kinase pathways—especially PI3K/Akt/mTOR and Ras/BRAF/Mek/Erk—were the most prevalent in the clinical and laboratory studies analyzing existing therapies and novel targets to treat adult-type diffuse glioma. (Table 2, Table 3 and Table 4) These results are consistent with previous studies that have demonstrated a predominance in the PI3K/Akt/mTOR and Ras/BRAF/Mek/Erk protein kinase pathways in molecularly targeted glioma treatment [5,6]. The importance of these pathways in glioma has been well-described in the literature; ultimately, these tumors harbor mutations that continuously activate these protein kinase signaling pathways, leading to increased tumorigenesis and progression [346,347,348].

Both the PI3K/Akt/mTOR and Ras/BRAF/Mek/Erk protein kinase pathways are also downstream of receptors such as EGFR, one of the most significant signaling pathways clinically implicated in glioma [349]. A systematic review of molecular targeted therapy clinical trials for GBM identified EGFR as the most prevalent molecular target [6]. Nonetheless, studies have demonstrated limited clinical benefit of anti-EGFR therapies, theorized to be secondary to PTEN-mediated resistance of GBM to this therapy type [350].

Similar to the published clinical studies on this topic, protein kinase pathways were by far the most predominant molecular targets tested in ongoing clinical trials. Interestingly, these therapeutics were also investigated much earlier on average. This finding is likely due to the fact that protein kinase inhibitors are some of the earliest molecular target therapies in the field of targeted oncologic interventions, thus being able to start clinical trials for the treatment of glioma as early as 2001 [351]. Perhaps, in the coming years, as the analysis of existing molecularly targeted therapies progresses from earlier stage clinical testing or laboratory testing, there will be a shift favoring more of the scientifically novel approaches—such as immunotherapeutics, cell cycle inhibitors, or more specifically localized targeting—in clinical trials.

Additional protein kinase pathways targeted in laboratory studies included HER2 receptors, epithelial membrane protein-2 (EMP2), and STAT3, to name a few [108,117,140]. HER2 expression tends to be low in GBM, and though one clinical trial examining a HER2 inhibitor has yet to show therapeutic gain, laboratory studies have promising evidence for efficacy [140,352]. EMP2 has been implicated in bevacizumab resistance and thus shows promise as a molecular target for preventing resistance in conjunction with this common therapeutic [117,353]. STAT3 plays a role in astrocyte development and has tumor suppressive roles in glial malignancies; this target shows promise in laboratory research using tetrandrine as an inhibitor [108]. Despite varying clinical evidence of efficacy, protein kinase-targeted therapies remain a prevalent area of study for both individual inhibitors and combined therapies.

### 4.3. Cell Cycle/Apoptosis Pathways

Interestingly, a prevalent molecular target in laboratory studies—both testing existing therapies and identifying novel targets—were cell cycle/apoptosis pathways. This difference may be attributed to the fact that clinical studies tend to focus on targets with existing FDA-approved therapies or targets that are more well-established in the literature, while studies with the goal of establishing new targets or testing newly developed therapies can explore a wider range of targets with less established evidence.

The use of cell cycle or apoptosis pathways as targets stems from the use of these pathways in the treatment of other tumors, in particular. In the present study, only four clinical studies included cell cycle/apoptosis pathway inhibitors, namely the cyclin-dependent kinase (CDK) 4/6 inhibitor palbociclib, the mouse double minute 2 (MDM2) inhibitor idasanutlin, the ribonucleotide reductase inhibitor Motexafin Gadolinium, and the 26S proteasome inhibitor bortezomib [36,57,58,59]. CDK and MDM2 inhibitors were also prevalent in laboratory studies testing existing therapies [150,156,160,161,165,176]. Two CDKs were identified as novel molecular targets—namely CDK 5 and 10—and other novel targets include other apoptosis regulators such as E2F1, trichothiodystrophy group A protein (TTDA), and protease activated receptor 2 (PAR2) [267,269,274,279,280].

### 4.4. Microenvironmental Pathways (Angiogenesis, Cell-Cell Adhesion, Ιron/Cation Regulation)

Anti-angiogenic therapies aim to compensate for the robust vascularity of gliomas, particularly GBM [354]. Specifically, vascular endothelial growth factor (VEGF) is overexpressed in GBM, providing rationale for the thirteen published clinical trials investigating VEGF inhibitors. Specific inhibitors studied include cediranib, cabozantinib, apatinib, and bevacizumab, which appear to be well-tolerated by patients and, in many cases, portend progression-free survival [32,39,40,41,42,43,45,46,48,49,50,55]. Many laboratory studies also tested VEGF inhibitors—namely bevacizumab, axitinib, and apatinib—and all found promising results in vivo [187,191,203]. Other microenvironmental targets included mitochondrial transcription factor A (TFAM), transient receptor potential cation channel subfamily V member 4 (TRPV4), and HIF2α. These targets were acted on by melatonin, cannabidiol, and PT2385, respectively, all of which demonstrated antitumor effects [183,186,198]. Promising novel microenvironmental targets include miR-497, TWIST transcription factor, and tenascin-W, among others [291,293,296].

### 4.5. Immunotherapy Pathways

The immune checkpoint blockade adopted in the glioma therapeutics model follows treatment paradigms for melanoma, lung cancer, colon cancer, and hepatocellular carcinoma; the therapies used to treat these tumors tend to block programmed cell death protein 1 (PD1), a protein known for attenuating the host immune response to tumor cells, or cytotoxic T lymphocyte antigen-4 (CTLA-4), a molecule that inhibits T-cell activation [355,356,357,358]. The clinical studies identified in the present study investigating immunotherapeutic pathways targeted PD1 using the inhibitor nivolumab [54,359]. Other immunotherapy targets that were found to be effective in vivo included the inhibition of CD73 with antibodies, extracellular matrix metalloproteinase (EMMPRIN) with icaritin, and NFκB with BAY117082 [207,210,211]. Novel immunotherapeutics for GBM and oligodendroglioma include cluster of differentiation 204 (CD204), S100A, and the CE7 epitope of the L1-CAM adhesion molecule [92,306,308].

### 4.6. Wnt/β-Catenin Pathway

The wnt/β-catenin pathway was much more prevalent in earlier stages of laboratory research identifying new targets, likely because the role of wnt/β-catenin in glioma progression is a more recent scientific advancement [310,311,312,313,314,315,316,349]. It is likely that in the upcoming years, the distribution of molecular targets may shift from protein kinase pathway-targeted therapies towards the wnt/β-catenin pathway or a combinatory approach of the two. Ongoing clinical trials have yet to target these pathways, but it is likely that this will soon change.

### 4.7. Study Design

The majority of laboratory studies utilized GBM cell lines or GBM patient samples. The frequent use of the U87 cell line in laboratory studies may be attributed to its widely accepted use as a model for GBM [360]. The use of technology such as spheroid or 3D cell culture is highly relevant in the context of therapies for gliomas. These technologies more accurately represent the tumor microenvironment and allow for better design of patient-specific treatments. Nearly one-third of laboratory studies testing existing therapies utilized this technology, implying that these studies are likely closer to translation to human studies.

The use of patient-derived GBM and glioma samples also highlights the importance of personalized medicine approaches in glioma treatment; nonetheless, this use also limits the generalizability of the conclusions, as most of these studies did not investigate molecular subtypes.

### 4.8. Implications

Molecular targeted therapy is predicted to revolutionize glioma therapy [361,362,363]. Particularly looking at the NCT Neuro Master Match (N2M2) trial, which uses molecular signatures of GBM to inform treatment, future studies will likely use the molecular identities of tumors to designate treatment [36]. These findings portend a shift in molecular targeted therapy research as well, wherein laboratory studies testing existing treatments will enter Phase I/II clinical trials and studies identifying novel targets will advance into the development and testing of therapies in a laboratory setting. Specifically, we will likely see a broadening of the current clinical studies and ongoing clinical trials—including more immunotherapeutics and microenvironmental pathway testing—in addition to testing of wnt/β-catenin pathway inhibitors in vitro and in vivo in the coming years.

### 4.9. Limitations

There are several limitations to our analysis that should be considered. First, there was significant heterogeneity in the patient populations, interventions, and outcomes reported across clinical studies. The quality of the studies included in our analysis also varied, with the majority having low levels of evidence due to being case reports or series. Notably, only 52 clinical studies were identified, which may be an underrepresentation of the true number of current clinical research studies investigating molecular targeted therapies for glioma. For instance, for GBM alone, a study analyzing the clinical trials related to molecular targeted therapy totaled 257 [6]. In contrast, the sum of published literature and ongoing clinical trials identified in this study totaled 171. This discrepancy is likely due to the fact that clinical trial titles may utilize specific drug names rather than the term “molecular targeted therapy” or broad names of categories within molecular targeted therapies.

Additional limitations include the fact that the studies had varying methodological quality and targeted different molecular pathways, making it difficult to draw definitive conclusions. The categorization of molecular targets is an imperfect model as well, for pathways such as STAT3 can simultaneously qualify as involving protein kinase inhibitors and angiogenesis, for instance [108]. The categorization of tumor types has also changed drastically since the WHO 2021 guideline change. This study retroactively reflects the updated tumor classification for these studies, using the literature to classify the mutation status of known cell lines. This may create a discrepancy between GBM literature released prior to 2021 and current models, but it more accurately reflects what these studies can add to future glioma literature. Our study, while comprehensive and broad in scope, is restricted by the vast variation, particularly in histological methodology and molecular marker identification capabilities.

Other limitations inherent to a systematic review are that of the search terms—for there may be studies about molecular targeted therapies that do not self-identify as such; publication bias from only including published studies; limiting the studies to only those available in English for full-text screen; and the lack of meta-analysis to quantify the data.

### 4.10. Future Directions

Future studies should aim to address these limitations by conducting larger multi-institutional clinical trials with standardized protocols and consistent reporting of outcomes. Studies should also consider investigating the effectiveness of combination therapies that target multiple molecular pathways simultaneously.

## 5. Conclusions

Here, we identify the current state of molecular target therapy research for adult-type diffuse gliomas, broadly found to be among one of three stages: validating molecular targeted therapies through published human clinical studies, testing existing therapies in a laboratory setting, and identifying novel molecular targets in a laboratory setting. We also queried clinicaltrials.gov for ongoing clinical trials on this topic. All studies predominantly investigated GBM, with few mentioning IDH-mutant astrocytomas or oligodendrogliomas. The most common molecular targets in all study types were protein kinase pathways such as PI3K/AKT/mTOR and Ras/BRAF/Mek/Erk. Microenvironmental targets were more numerous in clinical studies, whereas cell cycle/apoptosis were more numerous in laboratory studies. Immunotherapy pathways are few in number but on the rise in all study types, and the wnt/β-catenin pathway has been increasingly identified as a novel target.

Ultimately, these findings provide insight into the current state of molecular targeted therapy for glioma, highlighting the need for further investigation and the potential for this approach to improve patient outcomes.

## Figures and Tables

**Figure 1 ijms-24-10456-f001:**
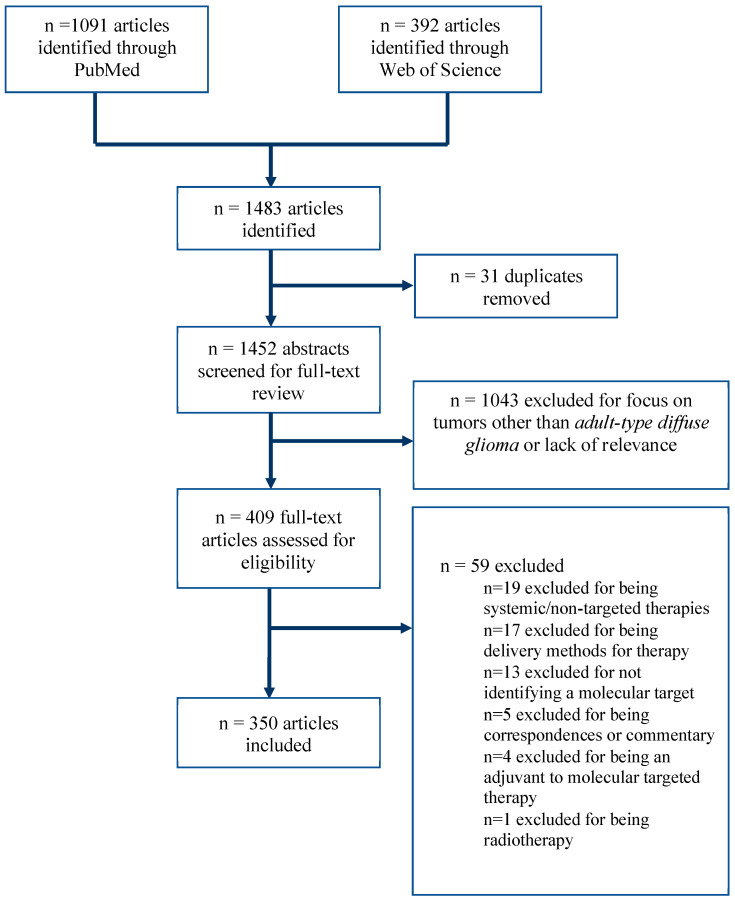
PRISMA flow diagram, demonstrating search pathway results and included articles.

**Figure 2 ijms-24-10456-f002:**
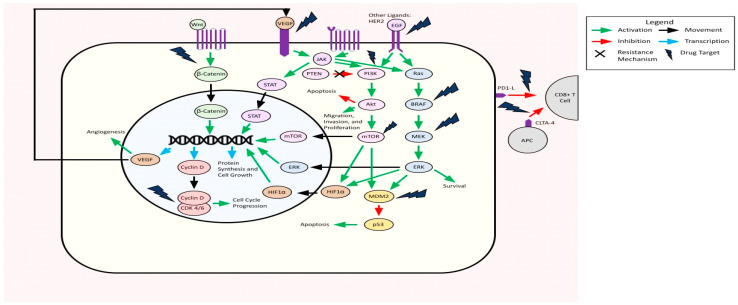
Summary of molecularly targeted pathways in adult-type diffuse glioma.

**Table 1 ijms-24-10456-t001:** Level of Evidence and Quality Assessment.

Level of Evidence (LoE)	Description
Level I	Evidence from a systematic review or meta-analysis of randomized control trials (RCTs) or evidence-based clinical practice guidelines based on RCTs.
Level II	Evidence obtained from at least one well-designed RCT (e.g., a large multi-site RCT).
Level III	Evidence obtained from well-designed controlled trials without randomization (i.e., quasi-experimental).
Level IV	Evidence from well-designed case-control or cohort studies.
Level V	Evidence from systematic reviews of descriptive and qualitative studies.
Level VI	Evidence from a single descriptive or qualitative study.
Level VII	Evidence from the opinions of authorities and/or reports of expert committees.

Level of effectiveness rating scheme adapted from Ackley et al. 2007 [12].

**Table 2 ijms-24-10456-t002:** Summary of clinical studies implementing molecular targeted therapies in glioma.

Study Author	Year	Tumor Type	Molecular Target	Intervention	Finding
Protein Kinase Pathways					
Berzero et al. [13]	2021	GBM, IDH-mutant Astrocytoma	RAF + MEK	Vemurafenib, Dabrafenib, Cobimetinib, Trametinib	The study highlights the long-term clinical benefits of RAFi/MEKi in adult patients with BRAF V600-mutant GGNTs
Butowski et al. [14]	2010	GBM	Protein kinase C-beta + PI3K/Akt	Enzastaurin + TMZ	Enzastaurin 250 mg/day given concomitantly with RT and temozolomide or adjuvantly with temozolomide was well tolerated
Chinnaiyan et al. [15]	2013	GBM	mTOR	Everolimus + TMZ + RT	Daily oral everolimus (10 mg) combined with both concurrent radiation and temozolomide, followed by adjuvant temozolomide, is well tolerated with an acceptable toxicity profile
Drobysheva et al. [16]	2017	IDH-mutant Astrocytoma	BRAF + MAPK	Dabrafenib + trametinib	PT1 and 2 were treated with MAPK and BRAF inhibitors and both showed marked responses, with PT1 only having a small residual abnormal signal at the primary tumor site and PT2 improving to stable disease
Franceschi et al. [17]	2012	GBM, IDH-mutant Astrocytoma	Src kinase	Dasatinib	Combination of CCNU and dasatinib showed significant hematological toxicities and led to suboptimal exposure to both agents
Fusco et al. [18]	2021	GBM, IDH-mutant Astrocytoma, Oligodendroglioma	BRAF + MEK	dabrafenib + Trametinib	Combination of BRAF/MEK inhibition has the potential to offer clinical benefit in both low-grade and high-grade gliomas
Hottinger et al. [19]	2019	Astrocytoma	MAPK + ERK	Dabrafenib + Trametinib	Reports and efficacy of dual BRAF/MEK inhibition in BRAF-mutated glioma
Johanns et al. [20]	2018	GBM	BRAF + MEK	Dabrafenib + Trametinib	PT1: 11mo therapy improved hemiparesis, speech, and functional status, after which the disease progressed and treatment was discontinued PT2: 3 mo therapy caused rapid response, allowing him to ambulate again, though he discontinued therapy and died shortly after
Kaley et al. [21]	2018	GBM, IDH-mutant Astrocytoma, and other gliomas	BRAF	Vemurafenib	BRAFv600 inhibition is a viable strategy, with a confirmed clinical benefit for 37.5% of patients and a best response of stable disease or better in 16/24 patients
Kanemaru et al. [22]	2019	Epithelioid GBM	BRAF + MEK	Dabrafenib and Trametinib	Dabrafenib and trametinib with radiation elicited a dramatic response in a patient with epithelioid GBM
Kebir et al. [23]	2019	GBM, IDH-mutant Astrocytoma	Multitarget kinase	Regorafenib	Study indicates a very poor performance of regorafenib in recurrent high-grade astrocytoma
Kleinschmidt-DeMasters et al. [24]	2015	GBM, IDH-mutant Astrocytoma	BRAF V600E kinase	Vemurafenib	E-GBMs can respond to targeted therapy
Lapointe et al. [25]	2020	GBM, IDH-mutant Astrocytoma	mTORC1/2	Vistusertib + TMZ	Combination of vistusertib with TMZ in GBM patients at first recurrence demonstrated a favorable safety profile at the tested dose levels
Lee et al. [26]	2012	GBM	Multitarget kinase + mTOR	Sorafenib + Temsirolimus	Minimal activity in recurrent glioblastoma multiforme was seen at the MTD of the two combined agents
Lombardi et al. [27]	2019	GBM	Multitarget kinase + mTOR	Regorafenib	REGOMA showed an encouraging overall survival benefit of regorafenib in recurrent GBM
Mason et al. [28]	2012	GBM	mTOR1	Everolimus + TMZ	Daily oral everolimus for 5 consecutive days every 28 days plus 150 mg/m^2^/day TMZ is an appropriate phase II dose for everolimus + TMZ
Migliorini et al. [29]	2017	Xanthoastrocytoma	BRAF + MEK	Dabrafenib + Trametinib	A patient with a refractory case of pleomorphic xanthoastrocytoma was treated with dual BRAF and MEK inhibition and exhibited a strong radiologic response
Rosenberg et al. [30]	2022	GBM, IDH-mutant Astrocytoma, and other gliomas	BRAF; BRAF + MEK	Vemurafenib + Dabrafenib + Trametinib	BRAF inhibition for BRAF-mutant glioma is a promising treatment paradigm; currently being evaluated prospectively in ACNS1723 clinical trial
Sanai et al. [31]	2018	GBM	Wee1K	AZD1775	AZD1775 reaches therapeutic concentrations in contrast-enhancing areas of GBM in humans and is well tolerated
Schiff et al. [32]	2015	GBM or anaplastic astrocytoma	MET and VEGFR2	Cabozantinib	Cabozantinib with TMZ and RT is well tolerated and warrants further evaluation
Shah et al. [33]	2007	Glioma	PDGFR	Imatinib and Hydroxyurea	Combining imatinib with hydroxyurea is effective in some glioma patients but is associated with dangerous myelosuppression
Shi et al. [34]	2019	IDH-wt, 1p19q co-deleted Glioma	BRAF V600E	Vemurafenib + Everolimus	Successfully treated a BRAF V600E-mutated anaplastic oligoastrocytoma with multiple extraneural metastases with vemurafenib and everolimus
Werner et al. [35]	2022	Glioma	Multitarget kinase	Regorafenib	Regorafenib is effective in recurrent grade III and IV gliomas, despite a high prevalence of level III and IV side effects
Wick et al. [36]	2019	GBM	ALK CDK4/6 mTOR MDM2 SHH	Alectinib Palbociclib Temsirolimus Idasanutlin Vismodegib	NCT Neuro Master Match (N2M2) trial Molecular signatures of GBM inform the treatment arm
Yau et al. [37]	2020	Ganglioglioma	BRAF + MEK	Vemurafenib and Cobimetinib	Combination BRAF and MEK inhibition is safe and feasible in a BRAF V600E unresectable ganglioglioma
Zustovich et al. [38]	2013	GBM	Multitarget kinase	Sorafenib	Combining sorafenib and temozolomide is feasible and safe and has activity in patients with relapsed GBM
Microenvironmental Targets (angiogenesis, cell-cell adhesion, iron/cation regulation)					
Badruddoja et al. [39]	2017	GBM	VEGF	Bevacizumab + TMZ	Bevacizumab plus bi-weekly temozolomide was well tolerated and may be a salvage regimen in recurrent glioblastoma
Brown et al. [40]	2016	GBM	VEGFR + EGFR	Cediranib + Gefitinib/placebo	Despite being underpowered with recruitment issues, this trial shows combining cediranib and gefitinib leads to increased PFS
Clarke et al. [41]	2014	GBM	VEGF + tyrosine kinase	Bevacizumab + Erlotinib	The combining of bevacizumab/erlotinib/ TMZ/radiotherapy appears to be well tolerated and improved progression-free survival but did not improve overall survival
D’Alessandris et al. [42]	2013	GBM	VEGF + EGFRvIII	Bevacizumab + Erlotinib	Obtained higher RR and PFS at 6 months (70%) than those reported in prior trials lacking molecular tumor analysis
Desjardins et al. [43]	2012	GBM	VEGF	Bevacizumab	Demonstrates that combined daily temozolomide and biweekly bevacizumab had some activity and was well tolerated
Hasselbalch et al. [44]	2010	GBM	EGFR, VEGF, topoisomerase I	Cetuximab + bevacizumab + irinotecan	None of the biomarkers tested alone or in combination could identify a patient population likely to benefit from bevacizumab and irinotecan, with or without the addition of cetuximab
Lassen et al. [45]	2015	GBM	Placental growth factor (PlGF) + VEGF	RO5323441 + Bevacizumab	Toxicity profile of RO5323441 plus bevacizumab was acceptable and manageable but not superior to bevacizumab alone
Lu et al. [46]	2014	GBM, Astrocytoma	VEGF	Bevacizumab + TMZ	After BEV treatment, most patients obtain more significant short-term responses with good toleration
Prados et al. [47]	2009	GBM or Gliosarcoma	EGFR	Erlotinib + TMZ + RT	Patients treated with erlotinib + TMZ + RT had improved survival
Vaccaro et al. [48]	2014	Glioma	VEGF	Bevacizumab	Bevacizumab and fotemustine showed anti-glioma activity and good tolerability among recurrent glioma patients
Vredenburgh et al. [49]	2012	GBM	VEGF	Bevacizumab + RT + TMZ	Addition of bevacizumab to the standard TMZ and RT regimen is associated with minimal toxicity
Wang et al. [50]	2014	GBM	EGFR	Nimotuzumab + TMZ + RT	Nimotuzumab, TMZ, and RT are safe therapeutic regimens, with similar survival times to other regimens
Wang et al. [51]	2017	Grade III and IV Glioma	VEGFR2	Apatinib + Irinotecan	Apatinib plus irinotecan is a potentially useful combination therapy and should be further evaluated
Weller et al. [52]	2017	GBM	EGFR	TMZ +/− Rindopepimut	Rindopepimut did not reduce mortality as a monotherapy in newly diagnosed GBM, so it may be necessary to use it in combination therapy
Wick et al. [53]	2020	Glioma	TGF β	TMZ+RT +/− galunisertib	There was no difference in safety or efficacy between the standard therapy and the standard plus galunisertib
Immunotherapy Pathways					
Anghileri et al. [54]	2021	GBM	PD1	Nivolumab	Nivolumab is useful for patients, despite a RCT failing to show overall benefits
Nayak et al. [55]	2021	GBM	PD1 + VEGF	Pembrolizumab + Bevacizumab	Pembrolizumab +/− bevacizumab is not an effective therapy
Reardon et al. [56]	2020	GBM	PD1	Nivolumab	Nivolumab monotherapy in GBM was equally safe and effective as bevacizumab monotherapy
Cell Cycle/Apoptosis/Transcription Pathways					
Brachman et al. [57]	2015	GBM or Gliosarcoma	Thioredoxin + ribonucleotide reductases	Motexafin Gadolinium + TMZ + RT	Combining standard RT with TMZ and MGd did not achieve a significant survival advantage
Kubicek et al. [58]	2009	GBM, Astrocytoma	26S Proteasome	Bortezomib	Bortezomib administered at its typical “systemic” dose (1.3 mg/m^2^) is well tolerated and safe in combination with TMZ and RT
Lin et al. [59]	2020	IDH-mutant Astrocytoma	CDK4	Palbociclib	First case of spinal cord tumor reported to demonstrate an association between CDK4 amplification and response to Palbociclib-based combination therapy even after multiple recurrences
Other					
Desjardins et al. [43]	2011	GBM	Farnesyl transferase	SCH 66336	The phase II dose of SCH 66336 when combined with standard 5-day temozolomide is 150 mg twice daily for patients on stratum A and 175 mg twice daily for patients on stratum B
Geletneky et al. [60]	2017	GBM	Protein NS1	Rat H-1 parvovirus (H-1PV)	Confirms H-1PV safety, tolerability and ability to cross the blood-brain barrier; favorable PFS compared with controls
Hashimoto et al. [61]	2015	GBM	WT1 (Wilms Tumor 1)	WT1 peptide vaccination + TMZ	Safety profile of the combined Wilms tumor 1 peptide vaccination and temozolomide therapy approach for treating glioblastoma was confirmed
Patel et al. [62]	2012	Glioma	ER	Tamoxifen + TMZ	The maximum tolerated dose of tamoxifen + TMZ + RT was 100 mg/m^2^
Sauter et al. [63]	2022	GBM	CSF1R, ABL, cKIT, PDGFR	Imatinib	Imatinib showed no effect on GBM

Abbreviation: RCT, randomized control trial; RAF, rapamycin associated factor; MEK, mitogen-activated protein kinase kinase; TMZ, temozolomide; PI3K/Akt, phosphoinositide-3-kinase/protein kinase B; GBM, glioblastoma multiforme; RT, radiation therapy; mTOR, mechanistic target of rapamycin; PT1; BRAF, v-raf murine sarcoma viral oncogene homolog B; MAPK, mitogen-activated protein kinase; CCNU, lomustine; ERK, extracellular signal-regulated kinase; e-GBM, epithelioid glioblastoma multiforme; MET, mesenchymal-epithelial transition factor; VEGFR, vascular endothelial growth factor receptor; PFS, progression-free survival; PT, patient; RR, response rate; CDK, cyclin-dependent kinase; MDM2, mouse double minute 2 homolog; SHH, sonic hedgehog; TGF-β, transforming growth factor-beta; PD1, programmed cell death protein 1; cSF1R, colony-stimulating factor 1 receptor; ABL, Abelson tyrosine-protein kinase 1; cKIT, receptor tyro-sine-protein kinase Kit; PDGFR, platelet-derived growth factor receptor.

**Table 3 ijms-24-10456-t003:** Summary of Laboratory Studies Implementing Molecular Targeted Therapies. All studies were level III evidence.

Study Author	Year	Tumor Type	Molecular Target	Intervention	Finding
Protein Kinase Pathways					
Aldea et al. [64]	2014	GBM	mTOR + RAF	Metformin + Sorafenib	Combining metformin and sorafenib is an effective treatment for TMZ-resistant glioblastoma cells
Aoki et al. [65]	2013	GBM	Ras	Nobiletin	Nobiletin inhibits Ras activity in C6 glioma cells
Arcella et al. [66]	2013	GBM	mTOR	Rapamycin	mTOR is upregulated in GBM and rapamycin represents a good inhibitor
Ariey-Bonnet et al. [67]	2020	GBM	MAPK14	BMZ	BMZ (Benzimidazole) is a potent inhibitor of MAPK14, which would directly contribute to its anticancer properties
Balkhi et al. [68]	2016	GBM	Multitarget kinases	Caffeic Acid Phenethyl Ester (CAPE) + Dasatinib	Combinational therapy inhibits migration and invasiveness and decreases cell survival
Barbarisi et al. [11]	2018	GBM	CD44	Quercetin + TMZ	CD44 targeted nanocarriers mediate site-specific delivery of quercetin via the CD44 receptor in GBM
Benezra et al. [69]	2012	GBM	Multitarget kinases	Dasatinib	Dasatinib has a significant survival benefit in vivo for mouse GBM
Camorani et al. [70]	2015	GBM	EGFRvIII	CL4 Aptamer + EGFR Tkis	CL4 and gefitinib cooperate with the anti-PDGFRβ Gint4.T aptamer in inhibiting cell proliferation.
Chen et al. [71]	2019	GBM	CD163 pathway (CK2, kinase)	TBB	By inhibiting CK2 with TBB (4,5,6,7-tetrabromo-1H-benzotriazole), it shows the CD163 pathway is crucial for tumor growth
Cheng et al. [72]	2022	GBM	CTSC	Piperlongumine + Scopoletin	CTSC (Cysteine cathepsin C) is a biomarker using the MAPK signaling pathway; inhibition with piperlongumine (more effective) and scopoletin decreases tumor growth
Ciesielski et al. [73]	2018	GBM	Src-kinase + tubulin polymerization inhibitory activity	Kx2-361	The drug is active in vivo against orthotopic GL261 gliomas in syngeneic C57BL/6 mice
Cloninger et al. [74]	2011	GBM	SAPK2/p38 + mTORC1	SB203580 + Rapamycin	Data support the combined use of SAPK2/p38 and mTORC1 inhibitors to achieve a synergistic antitumor therapeutic response
Combs et al. [75]	2007	GBM and Glioma	EGFR	Cetuximab	Triple combination of TMZ, RT, and cetuximab might be a promising multimodality treatment approach for patients with GBM
Dasgupta et al. [76]	2015	BRAF V600E GBM	BRAF V600E	Plx4720 + RT	Provide pre-clinical rationale for clinical trials of concurrent RT and BRAF V600E inhibitors
Dantas-Barbosa et al. [77]	2015	GBM and Ependymoma	mTOR	Γ-Secretase Inhibitor RO4929097	RO4929097, through mTOR inhibition, potentiates cytotoxicity in vitro but does not enhance antitumor effects in vivo
Davare et al. [78]	2018	GBM and other cell types	ROS1	Lorlatinib	ROS1 knockdown with lorlatinib resulted in powerful responses in mice
Di Stefano et al. [79]	2015	GBM	FGFR kinase	JNJ-42756493	JNJ-42756493 elicited potent growth inhibition and significant tumor regression after two weeks
Dominguez et al. [80]	2013	GBM	DGK-α	R59022 + R59949 + siRNA	DGK-α is a potential therapeutic glioma target linked to multiple key pathways
Du et al. [81]	2012	GBM	Raf/MEK/ERK signaling pathway	Sorafenib + Vitamin K (VK1)	Combining sorafenib with VK1 induced apoptosis through downregulating proapoptotic proteins Bcl-2 and Mcl-1
Emlet et al. [82]	2014	GBM	EGFRvIII + CD133	Egfrviii + CD133 AB	EGFRvIII + CD133 BsAb allow for the specific targeting of cancer stem cells
Farrell et al. [83]	2017	GBM	MET	WO2010/019899A1 + PF04217903 + Crizotinib	Dual targeting of HGF and MET by combining extracellular ligand inhibitors with intracellular MET TKIs could be an effective intervention
Feng et al. [84]	2010	GBM	PI3K/Akt; JNK; ERK	Tamoxifen	Mechanism of TAM-induced apoptosis reveal PI3K/Akt, JNK, and ERK as potential targets
Glassman et al. [85]	2021	GBM, Oligodendroglioma	MAPK kinase	U0126	Combining molecularly targeted therapies interferes more efficiently with glial tumor development and progression
Goker et al. [86]	2020	GBM	ALK	AZD3463 + TMZ	Combining TMZ with AZD3463 may increase the efficacy of a single TMZ treatment in GBM
Golubovskaya et al. [87]	2013	GBM	FAK	Y15	Blockade of FAK autophosphorylation with the oral administration of a small-molecule inhibitor, Y15, has the potential to be an effective therapy approach for GBM
Grossauer et al. [88]	2016	Glioma	BRAF/MEK	Dabrafenib + Trametinib	BRAF and MEK combination therapy helps to prevent MAPK reactivation during treatment
Gursel et al. [89]	2011	GBM and IDH-mutant Astrocytoma	PI3K/Akt	PI103/Pcn	PI-103 and TCN are sensitive inhibitors of the PI3K/Akt/mTOR pathway
He et al. [90]	2016	GBM	MEK2	MEK2 Antibody	MEK2 antagonists can be used as chemo-sensitizers to enhance the treatment efficacy of TMZ
Hjelmeland et al. [91]	2007	Astrocytoma	Raf + TOR	LBT613 + Everolimus	Combining LBT613 and RAD001 reduces glioma cell proliferation and invasion
Hong et al. [92]	2014	GBM	Aurora-A kinase	Alisertib	Inhibiting aurora-A kinase potentiatesthe effects of ionizing radiation on glioblastoma cells
Jiang et al. [93]	2018	GBM, other cell types	EGFR/EGFRvIII	EGFR/EGFRviii CAR T Cells	EGFR/EGFRvIII CAR T cells have strong anti-tumor and tumor-specific properties
Jin et al. [94]	2013	GBM	Akt + NOTCH	MRK003 + MK-2206	Akt and NOTCH inhibition decrease glioma proliferation
Joel et al. [95]	2015	GBM	PBK/TOPK	Hi-Topk-032	HITOPK-032 resulted in diminished tumor growth
Joshi et al. [96]	2012	GBM	Multitarget kinases	Gefitinib + Erlotinib + Sunitinib	Drug combinations containing sunitinib were most effective in vitro but not in vivo
Ju et al. [97]	2016	GBM	COX-2	Celecoxib	Targeting epirubicin plus celecoxib liposomes was able to effectively destroy the glioma VM channels and exhibited significant efficacy in glioma
Junca et al. [98]	2017	GBM	ALK, ROS1, MET	Crizotinib	MET and ALK are overexpressed in glioma; crizotinib is a potential molecularly targeted strategy
Jung et al. [99]	2014	GBM	FOXO3A	Z-Ajoene	Z-ajoene specifically targets glioma CSCs through the FOXO3A pathway
Kawauchi et al. [100]	2021	GBM	ALK	Alectinib + Ceritinib	Treatment with the second-generation ALK inhibitors, alectinib and ceritinib, might serve as a potent therapeutic strategy against GBM
Kim et al. [101]	2012	GBM, Astrocytoma	Phosphoinositide 3-kinase/Akt + Ras/Raf	5-Bromo-3-(3-Hydroxyprop-1-Ynyl)-2H-Pyran-2-One (BHP)	BHP targets GSCs and enhances their sensitivity to anticancer treatments
Koul et al. [102]	2005	GBM	Integrin-linked kinase	QLT0276 In DMSO	ILK inhibition down-regulates multiple pathways involved in proliferation and invasion
Koul et al. [103]	2010	GBM	PI3K/Akt	Px-866	PX-866 inhibits growth, induces G1 arrest and apoptosis, and is safe and effective in mouse models
Liu et al. [104]	2011	GBM	bFGF	Anti bFGF siRNA	bFGF (basic fibroblast growth factor) siRNA is a possible treatment for glioma
Liu et al. [105]	2014	GBM	EGFR and PI3K/Akt	G19	G19 acts on the EGFR and PI3K/Akt pathways and causes redox stress to kill glioma cells
Liu et al. [106]	2014	GBM	AMPK	Compound C	Compound C is an extremely potent antiglioma agent, though does not exclusively inhibit AMPK
Luchman et al. [107]	2014	GBM	mTOR1/2	AZD8055	Dual inhibition of mTOR1/2 with AZD8055 plus TMZ shows promise as a second-line treatment, especially in TMZ-resistant GBM
Ma et al. [108]	2015	GBM	STAT3	Tetrandrine	Tetrandrine inhibits glioma growth dose-dependently while not affecting the development of chick embryos
Matsuda et al. [109]	2012	GBM	JNK	Sp600125	JNK is involved in the development of stem-like potential in GBM cells and is an attractive target
Maxwell et al. [110]	2021	GBM	mTOR1/2 + MEK	TAK228 + Trametinib	Treatment with mTOR1/2 and MEK inhibitors induces various proteomic changes in gliomas
Nicolaides et al. [111]	2011	Astrocytoma	BRAF	Plx4720	BRAF inhibition as a treatment for astrocytoma is highly supported by preclinical findings
Paternot et al. [112]	2009	GBM	mTOR1 + MEK1/2	Rapamycin + PD184352	Combined inhibition of mTOR1 and MEK1/2 should be considered in tumors with dysregulated CDK4
Peng et al. [113]	2013	GBM	RACK1-PKC	siRNA	RACK1 is involved in glioma development via SRC/Akt activity
Pezuk et al. [114]	2013	GBM	PLK1	Bi2536 + Tmz	PLK1 is a promising molecular target, and inhibition + TMZ is effective in vitro
Phillips et al. [115]	2016	GBM and epidermoid carcinoma	EGFR	Abt-414	ABT-414 (antibody and MMAF fusion) is effective in treating a wide range of EGFR genotypes and can be advanced to phase I/II clinical trials
Premkumar et al. [116]	2010	GBM	IGF1R + Src	NVP-AEW541 + Dasatinib	Combined IGF1R and Src inhibition synergistically increased apoptosis in glioma cells without affecting normal astrocytes
Qin et al. [117]	2014	GBM	EMP2	Anti-EMP2 antibodies/Anti-EMP2 Igg1	EMP2 (epithelial membrane protein-2) promotes cell migration/invasion through protein kinases; inhibition kills tumor cells
Raub et al. [118]	2015	GBM	CDK4 + CDK6	Abemaciclib Or Palbociclib + TMZ	Ademacicib with TMZ synergistically increased rat survival time
Salphati et al. [119]	2012	GBM	PI3K	Gne-317	GNE-317 is a PI3K inhibitor designed to cross the blood brain barrier; represents a treatment option for GBM
Sathornsumetee et al. [120]	2006	GBM	BRAF, CRAF, VEGFR	AA1881	AAL881 treatment showed tumor growth retardation in xenograft tumors and was well tolerated by mice
See et al. [121]	2012	GBM	MEK + PI3K/mTOR	Vemurafenib + PI103	NF1-deficient GBM cell lines that are MEK inhibitor resistant respond well to dual therapy with MEK and PI3K/mTOR inhibition
Selvasaravanan et al. [122]	2020	GBM	MEK or PI3K	Trametinib + Pictilisib	MEK inhibition is not superior to PI3K inhibition, though MEK may have a use in combination therapy
Shingu et al. [123]	2015	GBM	MEK, EGFR, PI3K	Various Small Molecule Inhibitors	The most synergistic combinations of drugs affected RTKs and either MEK/ERK or PI3K
Siegelin et al. [124]	2010	GBM	BRAF	Sorafenib	sorafenib has potent in vivo and in vitro anti-glioma activity
Signore et al. [125]	2014	GBM	PDK1 + CHK1	UCN-01	UCN-01 downregulates PDK1 and CHK1, effectively killing tumor cells
Spino et al. [126]	2019	IDH-mutant Astrocytomas	DLL3	Rovalpituzumab Tesirine	DLL3 is selectively and homogeneously expressed in IDH-mutant astrocytomas and can be targeted with available MABs
Thanasupawat et al. [127]	2017	GBM	FGFR	Dovitinib	Alternation of dovitinib and TMZ reduces GBM viability independent of MGMT and p53 status
Thompson et al. [128]	2018	PXA	Various	Various Antibodies + Kinase Inhibitors + Chemo Drugs	Bevacizumab, TMZ, and irinotecan should be considered as adjuvant therapies for PXA, though MEK and TK inhibitors should be investigated as well
Tsigelny et al. [129]	2017	GBM	OLIG2	SKOG102	SKOG102 exhibited potent anti-glioma activity in vivo and in vitro by downregulating OLIG2
van den Heufel [130]	2017	PDX astrocytoma	MET	Compound A	Compound A prolonged survival of mice did not stop eventual progression
Wang J et al. [131]	2013	Glioma	MEK1	Mir-181b + TMZ	miR181b enhances the sensitivity of glioma cells to TMZ by downregulating MEK1
Wang et al. [132]	2014	GBM	RAS	Mir-143	miR-143 is downregulated in glioma and involved in the inactivation of RAS
Wang et al. [133]	2019	Glioma Stem Cells	EGFR or PI3K and DHODH	Lapatinib + BKM120 + Teriflunomide	Combined targeting of intrinsic synthetic enzymes reduces pyrimidine synthesis; presents an effective glioma paradigm
Wichmann et al. [134]	2015	GBM	EGFR and HER2	siRNA + Cetuximab + Trastuzumab	siRNA knock-down of EGFR and HER2 reduced the growth rate of GBM
Yan et al. [135]	2017	GBM	CSF-1R + cKIT + RTKs	PLX3397 + Vatalanib + Dovitinib	PLX3397 is an effective monotherapy and improves the efficacy of multiple tyrosine kinase inhibitors
Yang et al. [136]	2008	GBM	EGFR	Boronated EGFR MAB + Cetuximab	Both EGFR and EGFRvIII tumors must be targeted by a combination of boronated MAB and boronated cetuximab
Yao et al. [137]	2015	GBM	EGFR and BRAF	BRAF(V600E) Inhibitor PLX4720	Inhibiting EGFR and BRAF(V600E) decreased tumor cell proliferation, increased apoptosis, and extended survival
Zavalhia et al. [138]	2014	Ependymomas and oligodendromas	cKIT	Imatinib	C117+ tumors are susceptible to imatinib, and its use in their treatment should be further investigated
Zhang et al. [139]	2015	GBM	mGluR1	siRNA, Selective Antagonists Riluzole + BAY36-7620	Anti-tumor activity of mGluR1 inhibition in vivo was demonstrated
Zhang et al. [140]	2016	GBM	HER2	HER2 Specific NK Cells	Modified HER2-specific NK cells are effective against GBM
Zhang et al. [139]	2017	Glioma	BRAF V600E + MEK	PLX4032 + GDC0973	Combined BRAF V600E and MEK inhibition prevents tumor rebound by MAPK activation in glioma
Cell Cycle/Apoptosis/Transcription Pathways					
Bychkov et al. [141]	2020	GBM	S100A9 (one of the heterodimers for calprotectin)	shRNA	Mambalgin-2 inhibits glioma and GBM cells but not normal astrocytes
Chen et al. [142]	2013	GBM and Glioma Stem Cells	IGFBP3	IGFBP3 siRNA	S100A9 knockdown demonstrates a new anticancer strategy
Chen et al. [143]	2019	GBM	HDAC/EZH2	Compound 26/UNC1999	IGFBP3 depletion is a potential therapy through the induction of DNA damage and apoptosis
Grinshtein et al. [144]	2016	GBM	BAG3	BAG3 siRNA	HDAC and EZH2 inhibition in combination lead to synergistic effects in vitro
Festa et al. [145]	2011	GBM and IDH-mutant Astrocytoma	miR-27a (FOXO3a)	Antagomir-27a	BAG3 is highly expressed in gliomas; effective therapeutic target
Ge et al. [146]	2013	GBM	Tumor checkpoint controller targeting microtubules	BAL101553	MiR-27a may be up-regulated in human glioma, and antagomiR-27a of could inhibit proliferation and invasion ability
Genoud et al. [147]	2021	GBM	PAK5	PAK5 shRNA	BAL101553 is a promising therapeutic agent for glioblastoma and could synergize with innate immune stimulation
Gu et al. [148]	2015	GBM	DR4/5	TRAIL + Doxorubicin	PAK5 is overexpressed in glioma, and its inhibition blocks anti-apoptotic signals and promotes arrest
Guo et al. [149]	2011	GBM	CDK 4/6 + PDGFRα	Lenvatinib + Crenolanib + Abemaciclib + Palbociclib	TRAIL-LP and DOX-LP displayed stronger antiGBM effects than free drugs or liposomal drugs alone in vivo
Hamada et al. [150]	2022	Embryonic Kidney Cells	Procaspase-3	PAC-1 (*Activating Molecule)	Inhibitors targeting PDGFRα and CDK 4/6 signaling can treat patients with the p.K455_N468delinsN splice variant
Joshi et al. [96]	2017	GBM	Phospholipase C	D609	PAC1 + TMZ is feasible in a rodent model and a promising therapeutic regime
Kalluri et al. [151]	2017	Oligodendroglioma Stem Cells	NEK9	NEK9-siRNA	Chronic D609 treatment leads to decreased biomarker (Olig2) levels and G1 arrest
Kaneta et al. [152]	2013	GBM	BMI-1	Ptc-209	NEK9 inhibition causes spindle inhibition and mitotic catastrophe
Kong et al. [153]	2018	GBM	OPN	shRNA	Tumor growth is attenuated by PTC-2009; proof-of-concept for BMI-1 oncogene inhibition
Lamour et al. [154]	2015	GBM	PLK1	Bi2536	Tumorigenic potential of U87-MG sphere cells was completely abrogated upon OPN (osteopontin) silencing
Lee et al. [26]	2012	GBM	Wee1K	Mk-1775	PLK1 (polo-like kinase 1) is critical to survival of glioma cells; inhibition kills cells
Lescarbeau et al. [155]	2016	GBM	p53/MDM2	D-PMNIbeta	Wee1K phosphorylation is an effective anti-tumor target site
Li et al. [156]	2012	GBM	miR-23a (APAF1)	Anti-mir-23a	D-PMIBeta is an effective inhibitor of p53
Lian et al. [157]	2013	GBM	EGFR	AZD9291	miR-23a is upregulated in gliomas; knockdown reduces tumor survivability
Liu et al. [158]	2019	GBM	STK17A	Anti-STK17A shRNA	AZD9291 demonstrated efficient preclinical activity in GBM in vitro and in vivo models
Mao et al. [159]	2013	GBM	MDM2/4 + α5β1/αvβ3	Compound 9	STK17A portends a worse prognosis; knockdown reduces tumor survivability
Merlino et al. [160]	2018	GBM	CDK 4/6	PD-0332991	Compound 9 has the potential to be a potent anti-glioma therapy via MDM2/4 and α5β1/αvβ3 inhibition
Michaud et al. [161]	2010	GBM	FOXM1	Plumbagin	PD-0332991 inhibits glioma growth and increases survival
Niu et al. [162]	2015	GBM	XIAP + BCL-2	RIST + ARIST	Plumbagin significantly inhibited glioma cell proliferation and induced cell apoptosis
Nonnenmacher et al. [163]	2015	GBM	MGMT	PRIMA-1MET	RIST (rapamycin, irinotecan, sunitinib, and temozolomide) and aRIST (alternative to rapamycin, GDC-0941) prolonged survival time and reduced tumor burden
Patyka et al. [164]	2016	GBM and IDH-mutant Astrocytoma	MDM2	SP-141	p53 is the probable target of PRIMA-1MET, making it an effective targeted therapy.
Punganuru et al. [165]	2020	GBM	HSP90	BIIB021 + 17-AAG (HSP90 Inhibitor) + BRAFi +/Or MEKi	MDM2 inhibition by SP-141 can effectively curtail the growth of brain tumors in vitro and in vivo
Sasame J et al. [166]	2022	Embryonic Kidney Cells	HGFR/MET	Crizotinib	HSP90 inhibitor (plus BRAF or MEK inhibitors) overcome the limitations of current BRAFV600E mutant therapy
Tasaki et al. [167]	2016	IDH-mutant Astrocytoma and Glioma	IAPs	Gdc-0152	HGFR/MET is highly expressed in GSCs and could be inhibited by crizotinib
Tchoghandjian et al. [168]	2016	GBM	EGFR	Afatinib + TMZ	Inhibitors of apoptosis proteins (IAPs) are associated with lower survival rates, and GDC-0152 increases survival
Vengoji et al. [169]	2019	GBM	Survivin	Survivin-siRNA/Transferrin Receptor Conjugate	Afatinib plus TMZ significantly delayed progression and growth in vivo and in vitro
Wang et al. [170]	2011	GBM	EZH2	EZH2si-DMC	Conjugate decreases survivin expression and increases survival
Wang et al. [171]	2019	GBM	Carbamoyl-phosphate synthetase (CAD)	Teriflunomide	DMC nanoparticle-mediated EZH2-siRNA decreases tumor growth
Wang et al. [133]	2023	GBM	BCL6	RI-BPi	Targeting pyrimidine synthesis may yield an improved clinical outcome
Xu et al. [172]	2017	GBM and other cell types	CUL7	MIR-3940-5p	BCL6 is overexpressed in glioma and is associated with worse prognosis; RI-BPI reduces tumor growth
Xu et al. [173]	2020	Glioma	EGFRvIII	L8A4	CUL7 promotes tumorigenesis via NF-kappa B activation and can be negatively regulated by miR-3940-5p
Yang et al. [136]	2006	GBM	EF2-kinase	EF2-siRNA	Show the therapeutic efficacy of molecular targeting of EGFRvIII
Zhang et al. [174]	2011	GBM	ID2	Anti ID2 siRNA	EF2 (elongation factor 2) inhibits anoikis and regulates cell migration; knockdown inhibits these properties in tumor cells
Zhao et al. [175]	2015	GBM	CDK + Aurora (dual inhibitor)	Jnj-7706621	ID2 upregulation decreases apoptosis in glioma; targeting increases apoptosis and drug sensitivity
Zhong et al. [176]	2018	GBM and other cell types	ATG9A	Bevacizumab + Chloroquine	JNJ-7706621 was a potential drug for the treatment of patients with glioblastoma
Microenvironmental Targets (angiogenesis, cell-cell adhesion, iron/cation regulation)					
Abdul Rahim et al. [177]	2017	GBM	Phosphatidylserine	SAPC-DOPS	ATG9A depletion leads to cell death; however, chloroquine was found ineffective at non-toxic doses
Angara et al. [178]	2017	GBM	Endothelial pigpen protein	Aptamer III.1	HET0016 targets therapeutic resistance in glioma
Blanco et al. [179]	2014	GBM	NRP-1	NRP-1 Mab	SAPc-DOPS selectively targets GBM, crosses the BBB, and may be an effective treatment
Blank et al. [180]	2001	GBM	O-acetyl GD2 ganglioside	Anti-GD2 Antibody	Aptamer III.1 found to selectively target GBM and is a potential treatment
Chen et al. [181]	2013	GBM	TFAM	Melatonin + TMZ	NRP-1Mab is an inhibitor of glioma growth and invasion and may be an effective treatment
Fleurence et al. [182]	2016	GBM	Pan-VEGF	Cediranib + TMZ	O-acetyl GD2 ganglioside represents a new molecular target to prevent glioma proliferation
Franco et al. [183]	2018	GBM	LTβR	Light-VTP	Melatonin causes cell death and potentiates TMZ effects by inhibiting TFAM (mitochondrial transcription factor A)
Grossman et al. [184]	2013	GBM	TRPV4	Cannabidiol (CBD)	Intratumoral concentrations of TMZ in tumor ECF were slightly, but not statistically significantly, increased when compared to the treatment of TMZ alone
He et al. [185]	2018	GBM	VEGF + Src Family kinases	Bevacizumab + Dasatinib	LIGHT-VTP prevents angiogenesis, normalizes blood vessels, and promotes immune infiltration
Huang T et al. [186]	2021	GBM	Growth-Hormone Releasing Hormone	MIA-604 + MIA-690	Antitumor effect of CBD in glioma is caused by lethal mitophagy, and we identified TRPV4 as a molecular target
Huveldt et al. [187]	2013	GBM	Nrf2	siRNA	Dasatinib may block bevacizumab-induced invasion, and a phase II trial is being planned
Jaszberenyi et al. [188]	2013	GBM	MRP3	Anti-MRP Antibody	GHRH antagonists have potent anti-cancer activity, which can augment standard chemotherapeutic treatments
Ji et al. [189]	2013	GBM	VEGFR	Axitinib	Nrf2 promotes glioma proliferation and is inversely correlated with prognosis; siRNA may be a potential drug
Kuan et al. [190]	2010	GBM	TfR (transferrin receptor)	T12 + B6 + T7 (Tfr-Targeting Peptides)	MRP3 is overexpressed in gliomas; antibodies used in the study are specific to the tumors and decrease growth
Lu et al. [191]	2015	GBM and Glioma Stem Cells	CX43 + miR-21	B2 cAMP Agonist	Axitinib exhibits antiangiogenic activity and prolongs survival
Mojarad-Jabali et al. [192]	2022	GBM	Fibulin-3	Mab428.2	T7-modified liposomes (T7-LS) show BBB penetration capacity and demonstrate in vitro effectiveness
Mostafavi et al. [193]	2015	GBM and IDH-mutant Astrocytoma	LAT1	BCH	CX43 and miR-21 modulation using B2 agonists is effective therapy for low- but not high-grade glioma
Nandhu et al. [194]	2018	GBM	NHE9	Gold NEPTT	mAb428.2 inhibited fibulin-3, reduced tumor growth, and extended survival
Nawashiro et al. [195]	2006	GBM and Glioma	Lanosterol synthase	Mi-2	LAT1 expression is inversely correlated with survival time, and BCH arrested growth and killed tumor cells
Pall et al. [196]	2019	GBM	HIF2α	PT2385	Gold nanoparticle-enabled photothermal therapy (NEPTT) crosses the BBB, delivers the gold nanoparticles, and kills tumor cells
Phillips et al. [197]	2019	DIPG and GBM	EDB-FN	Docetaxel-Loaded EDB-FN Specific Micelles	Characterized pathway of MI-2 (menin inhibitor), existing glioma treatment
Renfrow et al. [198]	2020	GBM	VEGF	Anti-VEGF AB + Nimustine	HIF2α is a reasonable therapeutic target; PT2385 is an efficacious anti-tumor agent
Saw et al. [199]	2021	GBM, IDH-mutant Astrocytoma, and other cell types	tmTNFa	Recombinant IL2 or dsDNA	EDB-FN (extra domain B fibronectin) is a useful biomarker and has antitumor efficacy
Takano et al. [200]	2003	GBM	CTL1 (choline transporter-like protein 1)	AMB4269951	Combination of antiangiogenic therapy with standard chemotherapy is a promising avenue for future therapy
Tyrinova et al. [201]	2018	Glioma	VEGFR2	Apatinib	tmTNFa is upregulated by rIL-2 or dsDNA, which helps to restore dendritic cell anti-tumor activity
Watanabe et al. [202]	2020	GBM	Calmodulin, EGFR, aromatase	W-13 + Gefitinib + Exemestane	Amb4269951 has significant antitumor effects in glioma and was also without significant weight loss
Xia et al. [203]	2022	GBM	ITGA9	miR-148a	Apatinib decreases tumor growth through the induction of ferroptosis via the VEGFR2/Nrf2/Keap1 pathway
Xiong et al. [204]	2019	GBM	STING	ASA404	Identified three existing miRNA-based chemicals for use as therapy
Xu et al. [205]	2019	GBM	CD73	Anti-CD73	miR-148a can suppress the malignant phenotype of GBM by targeting ITGA9
Immunotherapy Pathways					
Baehr et al. [206]	2017	GBM	ATX + LPA receptors	siRNA	ASA404, an inhibitor of STING (stimulator of interferon gene), demonstrates efficacy subcutaneously but has no relevant activity in orthotopic brain models
Goswami et al. [207]	2020	GBM	EMMPRIN	Icaritin	Propose a combination therapy to target CD73 plus blockade of PD1 and CTLA-4, suggesting anti-CD73 be tested
Merrill et al. [208]	2004	GBM and Glioma	NFkB	BAY117082 + MG132	CD155 is highly expressed in glioma, and PVS-RIPO is highly effective in vitro
Schleicher et al. [209]	2011	GBM	FPR	F2 Procyanidins	ATX and LPA receptor downregulation radio-sensitizes tumor cells
Xu et al. [210]	2015	GBM	CXCR4	POL5551 + MCR89	Icaritin inhibits the invasion and EMT of GBM cells by targeting EMMPRIN (extracellular matrix metalloproteinase)
Zanotto-Filho et al. [211]	2011	GBM and Glioma	Site-1 protease	PF-429242	NFkB inhibition helps defeat resistance mechanisms, decreases viability, and exhibits some toxicity
Zhang et al. [212]	2009	GBM and Glioma	CXCR4	Tetramethylpyrazine	F2 procyanidins downregulates FPR (formyl peptide receptor) causing a cytotoxic effect
Other Pathways/Targets					
Barone et al. [213]	2014	GBM	Lactate (monocarboxylate) transporters	ACCA	Higher POL5551 tumor concentrations are associated with better survival, improving in combination with VEGF antagonism
Caruana et al. [214]	2017	GBM	APLNR	MM54 Or MM193 (APLNR Antagonists)	PF-429242 decreases viability, increases apoptosis and inflammation, and downregulates lipid synthesis
Chen et al. [215]	2013	GBM	Nestin	Anti-Nestin IGG	Tetramethylpyrazine’s effect on gliomas comes through the inhibition of CXCR4
Chen et al. [216]	2021	GBM	EEF1A1 + RPL11	Puromycin + Doxorubicin + Daunorubicin + Mitoxantrone	circ-ITCH inhibits tumor progression by regulating the miR-106a-5p/SASH1 axis
Colen et al. [217]	2011	GBM	MALAT1	Nanocomplex Targeting MALAT1 + TMZ	ACCA (α-cyano-4-hydroxycinnamic acid) inhibits lactate transport and can be used to target brain tumors
Harford-Wright et al. [218]	2017	GBM	IDH1R132H	AGI-5198 (In Combo with HDACi)	Inhibition of APLNR (apelin G-protein coupled receptor) results in a significant reduction in tumor growth
Ishiwata et al. [219]	2011	GBM	hnRNP A1/B2	Β-Asarone	Downregulating nestin is associated with decreased glioma proliferation, growth, and migration
Jiang et al. [220]	2021	Glioma	CRM1	S109	Database analysis comparing glioma and normal tissue resulted in the identification of two target genes and four possible drugs for glioma treatment
Kim et al. [221]	2018	GBM	LPAR1/3	KI16425	Concurrent treatment of TMZ and nanocomplex-mediated silencing of MALAT1 has a survival benefit
Kim et al. [222]	2019	IDH-mutant Astrocytoma	Dynamin 2	Dynole 34-2 + Cydyn 4-36	AGI-5198 attenuates histone deacetylase inhibitor (HDACi) resistance and presents a potential therapy combination
Li et al. [223]	2018	GBM and other cell types	c-Myb	Telomestatin	β-Asarone blocks the invasion and epithelial-mesenchymal transition of glioma cells via inhibiting hnRNP A1/B2
Liu et al. [224]	2016	GBM and Glioma	miR-25	miR-25 Inhibitor (Cat. No. 4464084)	CRM1 is a novel molecular target; S109 inhibits the proliferation of tumor cells
Loskutov et al. [225]	2018	GBM	PRC2 + BET bromodomain proteins	JQ1 + I-BET	LPA signaling knockdown reduced tumor growth
Luwor et al. [226]	2019	GBM	eIF-5A, DHS, DOHH (both eIF-5A activators)	Gc7	Dynamin 2 inhibition via CyDyn 4-36 reduces tumor growth
Miyazaki et al. [227]	2012	GBM	TRAILR	Recombinant TRAIL + TMZ	Telomestatin impairs survival and growth via disrupting the c-myb protoconcogene
Peng et al. [228]	2019	GBM	EFTUD1	EFTUD1 shRNA	miR-25, through wnt signaling, may serve as a promising molecular target for the treatment of glioma
Piunti et al. [229]	2017	DIPG and Glioma	PFK1	Clotrimazole	Oncogenic properties of the histone point mutation H3K27M are reduced by inhibiting PRC2 and BET proteins
Preukschas et al. [230]	2012	GBM	YAP1	Nsc682769	eIF5-A is overexpressed in gliomas and its activator DHS represents a possible molecular target
Saito et al. [231]	2004	GBM	α7 nAChR	Rslurp-1	TMZ + TRAIL have a synergistic effect on survival while being safe in tumor-bearing rats
Saito et al. [232]	2014	GBM	A1CF + FAM224A	shRNA	EFTUD1 (elongation factor such as GTPase 1) is overexpressed in glioma, and its downregulation induces arrest and apoptosis
Sanzey et al. [233]	2015	GBM	DLL3	Rova-T	Clotrimazole inhibits PFK1 (phosphofructokinase 1) and increases survivability
Saunders et al. [234]	2021	GBM	Smoothened	Gdc-0449	NSC682769 represents a new YAP1 (yes-associated protein 1) inhibitor that decreases glioma growth and proliferation
Shulepko et al. [235]	2020	GBM	KIF11	Ipinesib	rSLURP-1 demonstrates antitumor activity through nAChR inhibition
Song Y et al. [236]	2019	GBM and other cell types	Brevican	Anti-Deglycosylated Brevican Peptide	A1CF/FAM224A/miR-590-3p/ZNF143 positive feedback loop regulates the malignant progression of tumor cells
Spino et al. [126]	2019	GBM and IDH-mutant Astrocytoma	miR-128	Ginsenoside Rh2	DLL3 (delta-like ligand 3) is selectively and homogeneously expressed in this tumor type; it is target with Rova-T (rovalpituzumab tesirine)
Tu et al. [237]	2017	GBM	14-3-3	siRNA	Smoothened is an effective prognostic biomarker, and GDC-0449 should be further evaluated as a potential drug
Venere et al. [238]	2015	GBM	IDH1R132H	Wm17	Inhibition of KIF11 (kinesin family member 11) stopped tumor growth, impeded tumor initiation, and prolonged survival
von Spreckelsen et al. [239]	2021	GBM	FTO	SPI1 Inhibitor DB2313	Deglycosylated Brevican is specific to high-grade gliomas; its knockdown by the BTP-7 peptide presents a new therapy
Wu et al. [240]	2011	GBM			Rh2 inhibits tumor proliferation via miR-128 upregulation
Yan et al. [241]	2013	GBM	mTOR + RAF	Metformin + Sorafenib	14-3-3 downregulation causes decreased glioma survival
Zhang et al. [242]	2021	IDH-mutant Astrocytoma	Ras	Nobiletin	WM17 is a novel mutant IDH1 inhibitor that inhibits cell migration but not proliferation
Zhang et al. [243]	2022	GBM, IDH-mutant Astrocytoma, Oligodendroglioma	mTOR	Rapamycin	FTO (fat mass and obesity-associated protein) is a novel prognostic indicator and decreases tumor burden

Abbreviations: RAF, rapidly accelerated fibrosarcoma; Ras, rat sarcoma; mTOR, mammalian target of rapamycin; MAPK, mitogen-activated protein kinase; GBM, glioblastoma multiforme; EGFR, epidermal growth factor receptor; SAPK2, stress-activated protein kinase 2; TMZ, temozolomide; RT, radiotherapy; ROS1, ROS proto-oncogene 1; FGFR, fibroblast growth factor receptor; MET, mesenchymal-epithelial transition factor; JNK, Jun N-terminal kinase; ERK, extracellular signal-regulated kinase; ALK, anaplastic lymphoma kinase; FAK, focal adhesion kinase; TCN, tetra-cycline; MEK, mitogen-activated protein kinase kinase; NOTCH, neurogenic locus notch homolog protein; PBK/TOPK, PDZ-binding kinase/T-lymphokine-activated killer cell-originated protein ki-nase; COX-2, cyclooxygenase-2; FOXO3A, forkhead box O3a; CSCs, cancer stem cells; ILK, integrin-linked kinase; bFGF, basic fibroblast growth factor; AMPK, adenosine mono-phos-phate-activated protein kinase; STAT3, signal transducer and activator of transcription 3; PKC, protein kinase C; PLK1, polo-like kinase 1; EGF1R, epidermal growth factor receptor 1; EMP2, epithelial membrane protein-2; LAT, linker for activation of T-cells; HIF, hypoxia-inducible factor; TWIST, twist family BHLH transcription factor; CD, cluster of differentiation; PFK, phos-phofructokinase; PDK, pyruvate dehydrogenase kinase; ARHGAP15, Rho GTPase-activating protein 15.

**Table 4 ijms-24-10456-t004:** Summary of Laboratory Studies Identifying Novel Molecular Targets.

Study Author	Year	Tumor Type	Molecular Target	Finding
Protein Kinase Pathways				
Chen et al. [244]	2021	GBM	ACTL6A	ACTL6A (actin-like 6A) knockdown inhibits tumor migration via suppressing the Akt pathway and increases sensitivity to TMZ
Edwards et al. [245]	2006	GBM	Phosphatidylinositol 3-kinase/Akt	Treatment of GBM cells with ILKAS can decrease ILK protein levels and downstream phosphorylation of the cell survival protein PKB/Akt on Ser473, the site specifically phosphorylated by ILK
Gabler et al. [246]	2019	BRAF V600E-mutated glioma	ETS1	Concomitant BRAFV600E and TERT promoter mutations synergistically support cancer cell proliferation and immortalization through ETS1 (e-twenty-six transcription factor)
Gu et al. [247]	2015	GBM	ITSN1S	ITSN1 (Intersectin1-S) contributes to glioma growth through the Raf/MEK/ERK pathway; overexpression correlates with higher grade gliomas
Hou et al. [248]	2015	GBM	PERK	PERK (PKR-like kinase) silencing decreases tumor cell viability and ATP/lactate production; decreases tumor formation capacity
Iqbal et al. [249]	2016	GBM	PIM	Combination PIM (Proto-oncogene serine/threonine-protein kinase) and PI3K inhibition may be an effective regimen in treating heterogeneous tumors
Keating et al. [250]	2010	Astrocytoma	Mer and Axl RTKs	Mer and Axl RTK inhibition is a novel method to improve apoptotic response and chemosensitivity in astrocytoma
Kim et al. [251]	2016	Glioma Stem Cells	MLK4	MLK4 regulates the mesenchymal identity of GSCs
Lerner et al. [252]	2015	GBM	PLK1	PLK1 inhibition is especially effective against CD133+ GBM cell subpopulations
Liu et al. [253]	2013	GBM	EF-2 kinase	Targeting EF-2 kinase can enhance the anti-glioma activity of TMZ
Liu et al. [254]	2015	Glioma	GCN5	GCN5 (general control of nucleotide synthesis 5) potentiates tumor proliferation and invasion via STAT3 and Akt signaling pathways
Mao et al. [159]	2013	GBM	STK17A	STK17A is a p53 target gene that is upregulated in GBM and associated with worse outcomes, while knockdown reduces proliferation, invasion, and migration
Martinez-Saez et al. [255]	2016	Glioma	peIF4E	peIF4E (eukaryotic translation initiation factor 4E), activated by the Ras-Raf-MAPK pathway, is an independent predictor of survival
Qin et al. [117]	2014	GBM	EMP2	EMP2 is an activator of Src and represents a potential molecular target for glioma therapy
Shoshan et al. [256]	1999	Oligodendroma	NG2 and PDGFRa	NG2 and PDGFRa are both overexpressed in oligodendromas and may represent molecular target
Sulzmaier et al. [257]	2016	GBM	RSK2	RSK2 serine/threonine-protein kinase is upregulated in glioma and is associated with decreased survival rates; knockdown reduces proliferation
Sun et al. [258]	2020	GBM	Nrf2	Nrf2 inhibition leads to increased oxidative stress and decreased Ras/Raf/MEK activity
Thanasupawat et al. [127]	2018	GBM	CTRP8	The CTRP8-STAT3 axis has strong anti-apoptotic properties involved in TMZ resistance
Tsuruta et al. [259]	2011	Glioma	PDGFRa and G-CSFR	Gliomas highly express PDGFRa (Platelet-derived growth factor receptor) and G-CSFR (colony stimulating factor receptor)
Wang et al. [133]	2019	GBM	Pyrimidine Synthesis Pathway	GSCs are vulnerable to inhibition of both the mutated enzyme and the rate-limiting (carbamoyl phosphate synthetase 2)
Yamanaka et al. [260]	2006	Glioma	DDR1	DDR1 (discoidin domain receptor tyrosine kinase 1) is associated with glioma proliferation and a worsened prognosis
Zhang et al. [261]	2016	GBM	YAP1/TAZ-BIRC5	The Hippo/YAP kinase pathway is abnormally activated by LATS downregulation and not affected by MST in glioma tissues
Zhang et al. [262]	2022	GBM	NDRG1 promoter	CW-type zinc finger 2 promotes the proliferation, invasion, migration, and EMT of glioma by regulating PTEN/PI3K/AKT signaling via binding to the N-myc downstream regulated gene 1 promoter (NDRG1)
Zhao et al. [263]	2016	GBM	PI3K/Akt and JNK	Combined inhibition of the PI3K p110β isoform and JNK may serve as a potent and promising therapeutic approach
Zhou et al. [264]	2005	GBM	FPR	FPR (Formyl Peptide receptor) acts through the JAK/STAT pathway and is highly expressed in GBM and other high-grade gliomas
Zhu et al. [265]	2014	GBM	Pyk2 or Orai1	SOCE (store-operated Ca2+ entry) is enhanced in gliomas, and knockdown by either Pyk2 (proline-rich tyrosine kinase 2) or Orai 1 inhibition can act as a novel approach
Zohrabian et al. [266]	2009	GBM	MEK and ROCK	Rho/ROCK signaling is involved in GBM cell migration and proliferation and represents an ideal target
Cell Cycle/Apoptosis/Transcription Pathways				
Abe et al. [267]	2019	Glioma	CDK5	CDK (cyclin-dependent kinase) 5 regulates lamellipodia and filopodia; blockade may decrease cell migration
Bai et al. [268]	2014	Glioma Stem Cells	TRF2	TRF2 (telomeric repeat binding factor 2) inhibition blocks tumor proliferation and increases survival
Bai et al. [269]	2020	GBM	TTDA	TTDA (trichothiodystrophy group A protein) is an upstream regulator of p53-mediated apoptosis and acts as an oncogene
Cai et al. [270]	2021	GBM	TRIM32	TRIM32 (tripartite motif protein 32) is overexpressed in glioma cells, and its knockdown decreases tumor growth and potentiates the TMZ response
Cao et al. [271]	2010	GBM and IDH-mutant Astrocytoma	14-3-3-protein	14-3-3 inhibition is associated with increased apoptosis, while 14-3-3 is upregulated in glioma cells
Chiang et al. [272]	2012	GBM	WOX1	WOX1 overexpression inhibits p53 mutant glioma cells independent of the intrinsic apoptosis pathway
Feng et al. [273]	2019	GBM	TRIM14	TRIM 14 (Tripartite motif-containing 14) tumor suppressor promotes EMT via ZEB2 (Zinc finger E-box-binding homeobox 2)
Godoy et al. [274]	2021	GBM	E2F1	E2F1 suppression is associated with decreased growth, increased apoptosis and susceptibility to radiation, and delayed differentiation
Kang et al. [275]	2019	GBM	lncRNA RP11-732M18.3	Inhibition of thelncRNA RP11-732M18.3, which promotes G1/S cell cycle transition, could provide a novel therapeutic target for glioma treatment
Kikuchi et al. [276]	2017	GBM	DEPDC1	DEPDC1 (DEP domain containing 1) induced apoptosis through NF-κβ signaling
Klose et al. [277]	2011	GBM	BMP7	BMP7 (Bone Morphogenetic Protein 7) is a potent tumor suppressor that induces G1/S cell cycle arrest via the BMP/TGF-β pathway
Lan et al. [278]	2020	GBM and other cell types	SNRPG	Downregulation of SNRPG (Small Nuclear Ribonucleoprotein Polypeptide G) induces cell cycle arrest and sensitizes tumor cells to TMZ by targeting Myc through a p53-dependent signaling pathway
Li et al. [279]	2018	GBM	CDK10	CDK10 overexpression is associated with the inactivation of snail-mediated EMT
Luo et al. [280]	2014	Glioma and GBM	PAR2	PAR2 (protease-activated receptor 2) is overexpressed in glioma cells and is involved in preventing apoptosis
Ma et al. [281]	2017	GBM	miR-96	miR-96 suppresses the PDCD4 (programmed cell death protein 4) tumor suppressor and is associated with increased tumor growth
Meuth et al. [282]	2008	GBM	TASK3	TASK1 and TASK3 (TWIK-related acid-sensitive K channel 3) are expressed in human glioma cells and are linked to glioma apoptosis
Tong et al. [283]	2019	GBM	YB-1	YB-1 (Y-box binding protein 1) facilitates resistance of glioma cells to TMZ by activating MDM2/p53 signaling
Wirsching et al. [284]	2014	GBM and Glioma	TB4	TB4 (thymosin beta 4) expression is correlated with glioma grade, and it modulates p53 and TGF-β
Yan et al. [285]	2014	Glioma	PRMT5	PRMT5 (protein arginine methyltransferase 5) is a protein arginine methyltransferase that is overexpressed in gliomas; attenuation leads to cell-cycle arrest
Yuan et al. [286]	2022	GBM	HSP27	HSP27 (heat shock protein 27) depletion promotes erastin-induced ferroptosis of tumor cells
Microenvironmental Targets (angiogenesis, cell-cell adhesion, cation regulation)				
Chung et al. [287]	2018	Glioma and GBM	EMP2	EMP2 is a biomarker for glioma differentiation and correlates with decreased survival
Bao et al. [288]	2016	GBM and IDH-mutant Astrocytoma	CAP1	CAP1 (adenylate cyclase-associated protein 1), a cytoskeleton regulator, significantly contributes to tumor proliferation, migration, and invasion
Haining et al. [289]	2012	Glioma	LAT1/4F2hc	LAT1/4F2hc amino acid transporter expression is correlated with proliferation, angiogenesis, and worsened outcomes
Ji et al. [189]	2013	GBM	Nrf2 and HIF1α	Nrf2 expression is directly correlated with HIF1α expression and is associated with worse outcomes
Kaur et al. [290]	2012	GBM	Cadherin-11	cadherin-11 is associated with increased glioma survivability and mobility
Lan et al. [291]	2014	GBM and other cell types	miR-497	Hypoxia-induced miR-497 is overexpressed in glioma and decreases glioma cell sensitivity to TMZ by inhibiting apoptosis
Li et al. [292]	2017	Glioma	miR-150	miR-150 modulates the HIF1α pathway and upregulates glycolysis in glioma cells
Li et al. [293]	2020	Glioma	TWIST	TWIST transcription factor could be a predictor of poor prognosis in glioma patients; it shows a correlation with microvascular density
Liu et al. [294]	2016	Glioma and GBM	XBP1	XBP1 (X-box binding protein 1) silencing reduces glioma cell viability and tumor formation capacity; it decreases glioma cell viability and ATP/lactate production
Ljubimova et al. [295]	2004	Glioma and Meningioma	Laminin-8	Laminin-8 expression is highly correlated with tumor grades and inversely correlated with survival time
Martina et al. [296]	2010	GBM, IDH-mutant Astrocytoma, Oligodendroglioma	Tenascin-W	Tenascin-W is overexpressed in brain tumors and not in normal tissue; it is a marker for glioma-associated blood vessels and stimulates angiogenesis
Okubo et al. [297]	2010	Glioma	LAT1	LAT1 (L-type amino acid transporter 1) expression corresponds with a higher density of microvessels in glioma
Pointer et al. [298]	2017	GBM	hERG	High hERG (human ether-à-go-go-related gene) potassium ion channel expression is correlated with decreased survival
Shi et al. [299]	2019	GBM	SLC2A1	LINC00174 promotes cell invasion, migration, and upregulated SLC2A1(solute carrier family 2 member 1)
Wu et al. [300]	2016	GBM	37LRP	37LRP (37-kDa laminin receptor precursor) is a novel glioma target whose downregulation by siRNA is associated with decreased growth, invasion, and proliferation
Immunotherapy Pathways				
Han et al. [301]	2019	Glioma	HVEM	Immune checkpoint molecule herpesvirus entry mediator (HVEM) is overexpressed and associated with poor prognosis
Hong et al. [92]	2014	GBM and other tumor types	L1-CAM	The CE7 epitope of the L1-CAM adhesion molecule on tumors may be amenable to targeting by CE7R T cells, making it a promising target for adoptive immunotherapy
Ku et al. [302]	2011	GBM	CHI3L1	CHI3L1 (Chitinase 3 like 1) contributes to glioma progression through invasion, resistance, and growth
Lou et al. [303]	2017	GBM	NUDT21	NUDT21(nudix hydrolase 1) is an upstream regulator of the NF-κB pathway and a potential molecular target for the MES subtype of GBM
Saito et al. [304]	2017	GBM	KIF-20A	KIF-20A (kinesin family member 20A) is highly expressed in glioma cells but not normal brain tissue; its suppression blocks proliferation and reduces cytokinesis
Xu et al. [305]	2020	Glioma	PARP9	PARP9 may serve as an unfavorable prognosis predictor for glioma
Yuan et al. [306]	2019	Glioma	CD204	CD204 contributes to dysfunction of T cells in glioma
Yuan et al. [307]	2022	GBM	BACH1	BACH1 (BTB Domain and CNC Homolog 1) attenuates the tumor-associated macrophage mediated immune response, therefore creating an immunosuppressive tumor environment
Zhang et al. [308]	2021	Oligodendroglioma and Glioma	S100A	Via databases, the S100A family was heavily involved in glioma immune infiltration and may represent an effective target
Zhu et al. [309]	2022	Glioma	PYGL	PYGL (Glycogen Phosphorylase L) can be used as a new biomarker and molecular target for evaluating the prognosis and immunotherapy of glioma
Wnt/β-catenin Pathways				
Chen et al. [310]	2021	Glioma	WTN5A	WNT5A gene, which expresses Wnt-5a, is overexpressed in gliomas; promotes EMT and angiogenesis
Di et al. [311]	2021	GBM	SPZ1, CXXC4 pathway	SPZ1 (Spermatogenic Leucine Zipper 1) stimulates glioma’s malignant progression via targeting CXXC4
Friedmann-Morvinski et al. [312]	2016	GBM	OPN	OPN (osteopontin) plays a role in dedifferentiating glioma cells
Guo et al. [313]	2020	GBM	FRAT1	FRAT1 (frequently rearranged in advanced T cell lymphomas-1) contributes to the tumorigenesis of glioma cells through wnt signaling
Lan et al. [314]	2015	GBM	PomGnT1	Forced overexpression of PomGnT1 (peptide-O-linked mannose beta-1,2-N-acetylglucosaminyltransferase 1) promotes tumor progression via activation of beta-catenin
Mizobuchi et al. [315]	2008	GBM	REIC/Dkk-3	REIC/Dkk-3 (reduced expression in immortalized cells /Dickkopf-related protein 3) is involved in Wnt-mediated apoptosis and is downregulated in glioma
Zhou et al. [316]	2015	GBM and IDH-mutant Astrocytoma	HOTAIR	High HOTAIR (HOX Transcript Antisense RNA) expression was associated with poor outcomes; depletion inhibits tumor cell migration/invasion
Other Pathways/Targets				
Borsics et al. [317]	2010	GBM	PRAF2	PRAF2 (rab acceptor 1 domain family, member 2) downregulation reduces the invasiveness of tumor cells
Cui et al. [318]	2019	GBM	RHPN1-AS1	Knockdown of RHPN1-AS1 inhibits the proliferation, migration, and invasion of tumor cells
Dong et al. [319]	2021	GBM	ANTXR1	miR-381-3p could repress malignant behaviors in glioma by modulating ANTXR1 (anthrax toxin receptor 1)
Feve et al. [320]	2014	GBM	13 different GPCRs	The transcriptome study shows 13 possible novel pathways that can be targeted by new drugs; refer to Table 1 of Feve et al., 2014 [320]
Han et al. [321]	2017	GBM	TAGLN2	TAGLN2 (Transgelin-2) plays a role in promoting the development of human glioma
Hou et al. [322]	2022	Glioma Stem Cells	CircASPM	CircASPM is up-regulated in glioma tissues and is correlated with tumor progression and poor prognosis
Huang et al. [323]	2020	GBM	GAS5-AS1	LncRNA GAS5-AS1 (growth arrest specific 5) inhibits glioma proliferation, migration, and invasion via miR-106b-5p/TUSC2 axis
Li et al. [324]	2011	GBM	DLL4-Notch	Combination therapy to block DLL4-Notch signaling may enhance the efficacy of VEGF inhibitors
Li et al. [325]	2014	GBM	miRNA network	There are 14 miRNAs and 5 pathways in the network that can represent glioma targets; refer to Figure 6A of Li et al., 2014 [325]
Li et al. [326]	2019	GBM	LINC00319	LINC00319 (long intergenic non-protein coding RNA 319) is an oncogenic factor for glioma tumorigenesis; knockdown arrests the cell cycle and induces apoptosis
Li et al. [327]	2021	GBM and other cell types	IGF2BP2	SUMOylation of IGF2BP2 (insulin-like growth factor 2 mRNA binding protein 2) regulated the OIP5-AS1/miR-495-3p axis to promote vasculogenic mimicry in tumor cells
Liu et al. [328]	2015	GBM and Glioma	miR-27b	miR-27b may promote glioma cell invasion through direct inhibition of Spry2 (sprouty homolog 2) expression
Liu et al. [329]	2022	GBM	LINC01094	LINC01094 promotes glioma progression by modulating miR-224-5p/CHSY1 axis
Miller et al. [330]	2017	GBM	JMJD6	JMJD6 (Jumonji Domain Containing 6) mediates tumor growth in vivo; targeting reduces glioma progression
Noorani et al. [331]	2020	GBM	147 druggable genes	Whole genome sequencing of human tumors identified 147 druggable targets for EGFR-mutant GBM, refer to Table S8 in Noorani et al., 2020 [331]
Qiu et al. [332]	2015	GBM and Glioma	FoxJ2	FoxJ2 (forkhead box J2) suppresses cell migration and invasion in glioma, so upregulating may be a strategy
Rose et al. [333]	2021	GBM and other tumor types	11 surface proteins	Shotgun proteomics identified 11 new potential targets for glioma therapy; refer to Figure 2A of Rose et al., 2021 [333]
Sanzey et al. [233]	2015	GBM	PFK1 and PDK1	Knockdown of PFK1 and PDK1, as well as some other glycolytic enzymes, acts an important enzyme in the metabolic escape pathways of GBM
Sharma et al. [334]	2016	IDH-mutant Astrocytoma	EZH2	EZH2 (enhancer of zeste homologue 2) and miRNA reactors act as biomarkers for tumor progression
Sun et al. [335]	2017	GBM	FOXP3/ARHGAP15	FOXP3 (forkhead box P3) and ARHGAP15 are both underexpressed in glioma tissues, and their absence plays a role in EMT
Visvanathan et al. [336]	2018	GBM and Glioma	METTL3	METTL3 (methyltransferase-like 3) preserves stem-cell-like capabilities in glioma cells and mediates SOX2 radiation salvage
Wang et al. [337]	2014	GBM	TIP-1	TIP1 (tax interacting protein 1) increases glioma invasion and angiogenesis; knockdown increases survivability
Wei et al. [338]	2014	GBM and Glioma	ADAR2	The ADAR2 (adenosine deaminases acting on RNA 2) alternative splicing variant is upregulated in glioma cells and may contribute to the malignancy of gliomas
Weigle et al. [339]	2005	GBM and IDH-mutant Astrocytoma	SOX11	SOX11 is highly and specifically expressed in glioma cells; it reactivates during tumorigenesis
Xin et al. [340]	2020	GBM	NFIA-AS2	NFIA-AS2 (nuclear factor I A antisense RNA2 gene) could be a novel biomarker and therapeutic target for glioma patients
Zhang et al. [341]	2022	GBM and Oligodendroglioma	ANXA1	ANXA1 is overexpressed in glioma tissues, plays a role in invasion and infiltration, and is an independent prognostic factor in glioma
Zhou et al. [342]	2021	GBM and Glioma	miR-190a-3p	miR-190a-3p contributes to glioma proliferation/migration and negatively regulates YOD1; can be suppressed by miR inhibition

Abbreviations: ILK, integrin-linked kinase; TERT, telomerase reverse transcriptase; MLK, mammalian sterile 20-like kinase; PLK, polo-like kinase; EF, elongation factor; STK, serine/threonine kinase; EMP, epithelial membrane protein; PDGFR, platelet-derived growth factor receptor; RSK, ribosomal S6 kinase; CTRP, C1q/TNF-related protein; GSCs, glioma stem cells; PI3K/Akt, phosphatidylinositol 3-kinase/protein kinase B; JNK, Jun N-terminal kinase; MEK, mitogen-activated protein kinase kinase; ROCK, Rho-associated protein kinase; CDK, cyclin-dependent kinase; EMP, epithelial membrane protein; LAT, linker for activation of T cells; L1-CAM, L1 cell adhesion molecule; PARP9, poly(ADP-ribose) polymerase family member 9; CD, cluster of differentiation; CXXC, cysteine-rich CXXC domain-containing protein; RHPN1-AS1, RHPN1 antisense RNA 1; PFK, phosphofructokinase; PDK, pyruvate dehydrogenase kinase; ANXA, annexin A.

**Table 5 ijms-24-10456-t005:** Summary of Ongoing Clinical trials Testing Molecular Targeted Therapies in Glioma.

Title	NCT #	Year Started	Last Update	Tumor Type	Molecular Target	Intervention
Protein Kinase Pathways						
Imatinib Mesylate in Treating Patients with Recurrent Malignant Glioma or Meningioma	00010049	2001	2018	Recurrent Malignant Glioma or Meningioma	multiple tyrosine kinases	imatinib
Gefitinib in Treating Patients with Newly Diagnosed Glioblastoma Multiforme	00014170	2001	2013	GBM	EGFR	Gefitinib
CCI-779 in Treating Patients with Recurrent Glioblastoma Multiforme	00016328	2001	2013	GBM or Gliosarcoma	mTOR	temsirolimus
Gefitinib in Treating Patients with Recurrent or Progressive CNS Tumors	00025675	2001	2018	GBM or Anaplastic Gliomas	EGFR	gefitinib
Erlotinib in Treating Patients with Solid Tumors and Liver or Kidney Dysfunction	00030498	2001	2013	Gliomas and Brain Metastases	EGFR	Erlotinib
Gefitinib and Radiation Therapy in Treating Patients with Glioblastoma Multiforme	00052208	2002	2020	GBM, Gliosarcoma	EGFR	Gefitinib
Imatinib Mesylate in Treating Patients with Gliomas	00039364	2002	2012	Glioma	Multiple tyrosine kinases	Imatinib
Erlotinib in Treating Patients with Recurrent Malignant Glioma or Recurrent or Progressive Meningioma	00045110	2002	2017	Glioma on EIADs	EGFR	erlotinib
Erlotinib and Temozolomide with Radiation Therapy in Treating Patients with Glioblastoma Multiforme or Other Brain Tumors	00039494	2002	2013	GBM or Gliosarcoma	EGFR	Erlotinib
A Phase II Exploratory, Multicentre, Open-label, Non-comparative Study of ZD1839 (Iressa) and Radiotherapy in the Treatment of Patients with Glioblastoma Multiforme	00238797	2003	2011	GBM	EGFR	Gefitinib
Imatinib Mesylate in Treating Patients with Recurrent Brain Tumor	00049127	2003	2019	Adult glioma	Multiple tyrosine kinases	Imatinib
Everolimus and Gefitinib in Treating Patients with Progressive Glioblastoma Multiforme or Progressive Metastatic Prostate Cancer	00085566	2004	2016	Progressive GBM	mTOR, EGFR	everolimus + gefinib
Erlotinib Compared with Temozolomide or Carmustine in Treating Patients with Recurrent Glioblastoma Multiforme	00086879	2004	2017	GBM	EGFR	erlotinib + carmustine + TMZ
Sorafenib in Treating Patients with Recurrent or Progressive Malignant Glioma	00093613	2004	2014	GBM	PDGFR	Sorafenib
Lapatinib in Treating Patients with Recurrent Glioblastoma Multiforme	00099060	2004	2014	Recurrent GBM	HER2, EGFR	lapatinib
GW572016 to Treat Recurrent Malignant Brain Tumors	00107003	2005	2018	GBM or gliosarcoma	EGFR/HER2	lapatinib
Ph I Gleevec in Combo w RAD001 + Hydroxyurea for Pts w Recurrent MG	613132	2005	2013	Recurrent Malignant GBM	multiple tyrosine kinases, mTOR	imatinib + RAD001 + hydroxyurea
Phase II Imatinib + Hydroxyurea in Treatment of Patients with Recurrent/Progressive Grade II Low-Grade Glioma (LGG)	00615927	2006	2013	Astrocytomas or oligodendromas	Multiple tyrosine kinases	Imatinib + hydroxyurea
Oral Tarceva Study for Recurrent/Residual Glioblastoma Multiforme and Anaplastic Astrocytoma	00301418	2006	2016	GBM and Anaplastic Astrocytoma	EGFRvIII	Erlotinib
Sorafenib Tosylate and Temsirolimus in Treating Patients with Recurrent Glioblastoma	00329719	2006	2018	Recurrent GBM	multiple kinases, mTOR	sorafenib + temsirolimus
Sorafenib Combined with Erlotinib, Tipifarnib, or Temsirolimus in Treating Patients with Recurrent Glioblastoma Multiforme or Gliosarcoma	00335764	2006	2018	GBM or Gliosarcoma	PDGFR, EGFR, farnesyltransferase, mTOR	Sorafenib, erlotinib, tipifarnib, and temsirolimus
Temsirolimus, Temozolomide, and Radiation Therapy in Treating Patients with Newly Diagnosed Glioblastoma Multiforme	00316849	2006	2013	GBM or Gliosarcoma	mTOR	temsirolimus + RT + TMZ
Tumor Tissue Analysis in Patients Receiving Imatinib Mesylate for Malignant Glioma	00401024	2006	2018	Glioma	Multiple tyrosine kinases	Imatinib
Erlotinib and Sorafenib in Treating Patients with Progressive or Recurrent Glioblastoma Multiforme	00445588	2007	2016	Recurrent GBM	ras-raf-MEK, mTOR	erlotinib + sorafenib
Dasatinib in Treating Patients with Recurrent Glioblastoma Multiforme or Gliosarcoma	00423735	2007	2019	GBM or Gliosarcoma	Multiple Kinases	Dasatinib
A Phase II Trial of Sutent (Sunitinib; SU011248) for Recurrent Anaplastic Astrocytoma and Glioblastoma	00606008	2007	2012	GBM or Anaplastic Astrocytoma	Multiple kinases	Sunitinib
Ph II Erlotinib + Sirolimus for Pts w Recurrent Malignant Glioma Multiforme	00672243	2007	2013	GBM	EGFR + IL2	Erlotinib + sirolimus
Radiation Therapy and Temozolomide Followed by Temozolomide Plus Sorafenib for Glioblastoma Multiforme	00544817	2007	2016	GBM	PDGR	Sorafenib + TMZ + RT
Sunitinib Tumor Levels in Patients Not on Enzyme-Inducing Anti-Epileptic Drugs Undergoing Debulking Surgery for Recurrent Glioblastoma	00864864	2007	2016	Recurrent GBM	multiple tyrosine kinases	sunitinib
Sunitinib in Treating Patients with Recurrent Malignant Gliomas	00499473	2007	2016	Recurrent Malignant Gliomas	multiple kinases	sunitinib
Ph. 2 Sorafenib + Protracted Temozolomide in Recurrent GBM	00597493	2007	2013	Recurrent GBM	PDGFR	Sorafenib + TMZ
Ph I Dasatinib + Erlotinib in Recurrent MG	00609999	2008	2014	Recurrent Malignant Glioma	multiple kinases, EGFR	dasatinib + erlotinib
Ph I SU011248 + Irinotecan in Treatment of Pts w MG	00611728	2008	2014	GBM	Multiple kinases	Sunitinib + Irinotecan
BIBW 2992 (Afatinib) with or without Daily Temozolomide in the Treatment of Patients with Recurrent Malignant Glioma	00727506	2008	2017	Recurrent Grade III and IV glioma	ErbB	Afatinib
A Study of Temsirolimus and Bevacizumab in Recurrent Glioblastoma Multiforme	00800917	2008	2010	recurrent primary GBM	mTOR, VEGF	temsirolimus + bevacizumab
Everolimus in Treating Patients with Recurrent Low-Grade Glioma	00823459	2009	2020	Low-Grade Glioma	mTOR	everolimus
Sorafenib in Newly Diagnosed High Grade Glioma	00884416	2009	2014	Newly Diagnosed High Grade Glioma	Multiple Kinases	sorafenib + TMZ + RT
Everolimus, Temozolomide, and Radiation Therapy in Treating Patients with Newly Diagnosed Glioblastoma	00553150	2009	2020	Grade IV gliomas	mTOR	everolimus + TMZ
Study of Sunitinib Before and During Radiotherapy in Newly Diagnosed Biopsy-only Glioblastoma Patients	01100177	2009	2013	GBM	Multiple kinases	Sunitinib
Dasatinib or Placebo, Radiation Therapy, and Temozolomide in Treating Patients with Newly Diagnosed Glioblastoma Multiforme	00869401	2009	2020	GBM	Multiple kinases	TMZ + RT +/− dasatinib
Open Label Trial to Explore Safety of Combining Afatinib (BIBW 2992) and Radiotherapy with or without Temozolomide in Newly Diagnosed Glioblastoma Multiform	00977431	2009	2019	GBM	EGFR	afatinib + RT + TMZ
Radiation Therapy and Temsirolimus or Temozolomide in Treating Patients with Newly Diagnosed Glioblastoma	01019434	2009	2018	GBM	mTOR	temsirolimus + TMZ
Temsirolimus and Perifosine in Treating Patients with Recurrent or Progressive Malignant Glioma	01051557	2010	2021	Glioma	mTOR	perifosine + temsirolimus
A Study in Subjects with Recurrent Malignant Glioma	01137604	2010	2022	Recurrent Malignant Gliomas	multiple tyrosine kinase inhibitor, VEGF	lenvatinib + bevacizumab
Bafetinib in Treating Patients with Recurrent High-Grade Glioma or Brain Metastases	01234740	2010	2018	Glioma or brain met	ABL1	Bafetinib
Everolimus, Temozolomide, and Radiation Therapy in Treating Patients with Newly Diagnosed Glioblastoma Multiforme	01062399	2010	2022	Newly Diagnosed GBM	mTOR	everolimus + RT + TMZ
EGFR Inhibition Using Weekly Erlotinib for Recurrent Malignant Gliomas	01257594	2011	2023	Glioma	EGFR	Erlotinib
AZD8055 for Adults with Recurrent Gliomas	01316809	2011	2019	Recurrent gliomas	mTOR	AZD8055
Phase I-II Everolimus and Sorafenib in Recurrent High-Grade Gliomas	01434602	2012	2022	GBM or anaplastic gliomas	mTOR1/2 + PDGFR	everolimus + sorafenib
Lapatinib with Temozolomide and Regional Radiation Therapy for Patients with Newly-Diagnosed Glioblastoma Multiforme	01591577	2012	2022	Newly-Diagnosed GBM Multiforme	EGFR	lapatinib + TMZ
Sorafenib, Valproic Acid, and Sildenafil in Treating Patients with Recurrent High-Grade Glioma	01817751	2013	2023	Recurrent High-Grade Glioma PDGFRa+	PDGFRA	kinase inhibitor Sorafenib + Valproic acid + Sildenafil
Lapatinib Ditosylate Before Surgery in Treating Patients with Recurrent High-Grade Glioma	02101905	2014	2023	EGFR Amplified Recurrent High-Grade Glioma	EGFR	Lapatinib
Study to Evaluate Safety and Activity of Crizotinib with Temozolomide and Radiotherapy in Newly Diagnosed Glioblastoma	02270034	2014	2022	GBM	ALK	crizotinib
Perifosine and Torisel (Temsirolimus) for Recurrent/Progressive Malignant Gliomas	02238496	2014	2023	Recurrent Glioma	mTOR	Temsirolimus, Perifostine
Study of LY2228820 with Radiotherapy Plus Concomitant TMZ in the Treatment of Newly Diagnosed Glioblastoma	02364206	2015	2019	GBM	p38 MAPK	LY2228820
Study of Tesevatinib Monotherapy in Patients with Recurrent Glioblastoma	02844439	2016	2021	GBM	EGFR, VEGFR, HER2	Tesevatinib
Dabrafenib and/or Trametinib Rollover Study	03340506	2017	2023	High Grade Glioma	B-Raf, MEKi	dabrafenib + trametinib
Ruxolitinib with Radiation and Temozolomide for Grade III Gliomas and Glioblastoma	03514069	2018	2023	Grade III Gliomas and GBM	JAK/STAT	Ruxolitinib + RT +TMZ
A Trial of Ipatasertib in Combination with Atezolizumab	03673787	2018	2022	GBM	AKT, PD-L1	Ipatasertib, Atezolizumab
18F-FDG PET and Osimertinib in Evaluating Glucose Utilization in Patients with EGFR Activated Recurrent Glioblastoma	03732352	2018	2023	EGFR Activated Recurrent GBM	EGFR	Osimertinib
9-ING-41 in Patients with Advanced Cancers	03678883	2019	2023	Malignant glioma	GSK-3β	9-ING-41
Nedisertib and Radiation Therapy, Followed by Temozolomide for the Treatment of Patients with Newly Diagnosed MGMT Unmethylated Glioblastoma or Gliosarcoma	04555577	2020	2022	Newly Diagnosed MGMT Unmethylated GBM or Gliosarcoma	DNA-dependent protein kinase (DNA-PK)	Nedisertib + RT
Tofacitinib in Recurrent GBM Patients	05326464	2022	2023	Recurrent GBM	JAK	Tofacitinib
DETERMINE Trial Treatment Arm 5: Vemurafenib in Combination with Cobimetinib in Adult Patients with BRAF Positive Cancers.	05768178	2023	2023	Glioma	BRAF V600	Vemurafenib + Cobimetinib
Superselective Intra-arterial Cerebral Infusion of Temsirolimus in HGG	05773326	2023	2023	recurrent high-grade glioma (grade 3 or 4 per WHO criteria)	mTOR	Temsirolimus
Microenvironmental Targets (angiogenesis, cell-cell adhesion, iron/cation regulation)						
Gefitinib Plus Temozolomide in Treating Patients with Malignant Primary Glioma	00027625	2002	2018	Malignant Primary Glioma	EGFR	gefitinib + TMZ
Safety and Efficacy Study of Tarceva, Temodar, and Radiation Therapy in Patients with Newly Diagnosed Brain Tumors	00187486	2004	2017	GBM or Gliosarcoma	EGFR	erlotinib + TMZ
Erlotinib and Temsirolimus in Treating Patients with Recurrent Malignant Glioma	00112736	2005	2015	Recurrent Malignant Glioma	EGFR, mTOR	erlotinib + temsirolimus
Temozolomide and Radiation Therapy with or without Vatalanib in Treating Patients with Newly Diagnosed Glioblastoma Multiforme	00128700	2005	2012	GBM	VEGFR	vatalanib + TMZ
Imatinib Mesylate, Vatalanib, and Hydroxyurea in Treating Patients with Recurrent or Relapsed Malignant Glioma	00387933	2005	2015	Recurrent or Relapsed Malignant Glioma	VEGF, multiple tyrosine kinases	imatinib + vatalanib + hydroxyurea
Cetuximab, Bevacizumab and Irinotecan for Patients with Malignant Glioblastomas	00463073	2006	2008	Malignant GBM	VEGF, EGFR	bevacizumab + cetuximab + irinotecan
PTK787/ZK 222584 in Combination with Temozolomide and Radiation in Patients with Glioblastoma Taking Enzyme-Inducing Anti-Epileptic Drugs	00385853	2006	2013	GBM	VEGF	PTK787/ZK (volitinib) + TMZ + RT
Pazopanib In Combination with Lapatinib in Adult Patients with Relapsed Malignant Glioma	00350727	2006	2013	Recurrent Glioma	VEGFR, HER2	pazopanib and lapatinib
Phase (Ph) II Bevacizumab + Erlotinib for Patients (Pts) with Recurrent Malignant Glioma (MG)	00671970	2007	2013	Recurrent Malignant Gliomas	EGFR, VEGF	erlotinib + bevacizumab
Bevacizumab and Cediranib Maleate in Treating Patients with Metastatic or Unresectable Solid Tumor, Lymphoma, Intracranial Glioblastoma, Gliosarcoma or Anaplastic Astrocytoma	00458731	2007	2014	Metastatic GBM, Gliosarcoma, or Anaplastic Astrocytoma	VEGF	bevacizumab + cediranib maleate
Study of Bevacizumab Plus Temodar and Tarceva in Patients with Glioblastoma or Gliosarcoma	00525525	2007	2014	GBM or Gliosarcoma	VEGF + EGFR	bevacizumab + erlotinib + TMZ
Ph I Zactima + Imatinib Mesylate and Hydroxyurea for Pts w Recurrent Malignant Glioma	00613054	2007	2012	Recurrent Malignant Glioma	VEGFR, PI3KT, EGFR, PDGFR	Zactima + imatinib + hydroxyurea
Cediranib, Temozolomide, and Radiation Therapy in Treating Patients with Newly Diagnosed Glioblastoma	00662506	2008	2017	GBM or Gliosarcoma	VEGFR	Cediranib
Bevacizumab and Sorafenib in Treating Patients with Recurrent Glioblastoma Multiforme	00621686	2008	2018	Recurrent GBM	VEGF + multiple tyrosine kinases	bevacizumab + sorafenib
RT, Temozolomide, and Bevacizumab Followed by Bevacizumab/Everolimus in First-line Treatment of GBM	00805961	2009	2021	GBM	VEGF, mTOR1/2	Bevacizumab + Everolimus + RT + TMZ
Afatinib (BIBW 2992) QTcF Trial in Patients with Relapsed or Refractory Solid Tumours	00875433	2009	2013	Relapsed or Refractory Solid Tumours (GBM and brain metastases)	EGFR	Afatinib
Bevacizumab and Erlotinib After Radiation Therapy and Temozolomide in Treating Patients with Newly Diagnosed Glioblastoma Multiforme or Gliosarcoma	00720356	2009	2018	GBM	VEGF + EGFR	Bevacizumab + Erlotinib
Dasatinib and Bevacizumab in Treating Patients with Recurrent or Progressive High-Grade Glioma or Glioblastoma Multiforme	00892177	2009	2019	GBM	VEGF + multiple kinases	Bevacizumab + dasatinib
Temozolomide and Radiation Therapy with or without Cediranib Maleate in Treating Patients with Newly Diagnosed Glioblastoma	01062425	2010	2022	GBM	VEGFR	TMZ + RT +/− cediranib
Cediranib Maleate and Cilengitide in Treating Patients with Progressive or Recurrent Glioblastoma	00979862	2010	2015	GBM or Gliosarcoma	VEGFR, integrins	Cediranib and Cilengitide
Gamma-Secretase Inhibitor RO4929097 and Cediranib Maleate in Treating Patients with Advanced Solid Tumors	01131234	2010	2014	Gliomas and Brain Mets	VEGFR and gamma secretase	Cediranib + RO4929097
A Study of Avastin (Bevacizumab) and Irinotecan Versus Temozolomide Radiochemistry in Patients with Glioblastoma	00967330	2010	2015	Newly diagnosed GBM, non-methylated MGMT promoter	VEGF	bevacizumab + irinotecan + TMZ + RT
BIBF 1120 in Recurrent Glioblastoma Multiforme	01251484	2011	2012	Recurrent GBM	VEGFR	Cediranib
BIBF 1120 for Recurrent High-Grade Gliomas	01380782	2012	2014	GBM or Anaplastic Gliomas	VEGFR/PDGFR/FGFR	Nintedanib
CAR T Cell Receptor Immunotherapy Targeting EGFRvIII for Patients with Malignant Gliomas Expressing EGFRvIII	01454596	2012	2019	Malignant Gliomas Expressing EGFRvIII	EGFRvIII	CAR T cell targeting EGFRvIII
Tivozanib for Recurrent Glioblastoma	01846871	2013	2019	GBM	VEGFR	Tivozanib
A Randomized Phase II Clinical Trial on the Efficacy of Axitinib as a Monotherapy or in Combination with Lomustine for the Treatment of Patients with Recurrent Glioblastoma	01562197	2014	2019	GBM	VEGFR	Axitinib
Apatinib in Recurrent or Refractory Intracranial Central Nervous System Malignant Tumors	03660761	2016	2019	GBM	VEGFR2	Apatinib + TMZ
Safety Study of Afatinib for Brain Cancer	02423525	2016	2022	Recurrent or Progressive Brain Cancer	VEGF	afatinib
Clinical Trial on the Combination of Avelumab and Axitinib for the Treatment of Patients with Recurrent Glioblastoma	03291314	2017	2019	Recurrent GBM	VEGFR, PD1L	Axitinib + Avelumab
Prediction of Therapeutic Response of Apatinib in Recurrent Gliomas	04216550	2018	2021	Recurrent Gliomas	VEGFR-2	Apatinib
Ketoconazole Before Surgery in Treating Patients with Recurrent Glioma or Breast Cancer Brain Metastases	03796273	2019	2022	Recurrent Glioma or Breast Cancer Brain Metastases	tGLI1	ketoconazole
Anlotinib Combined with STUPP for MGMT Nonmethylated Glioblastoma	04725214	2021	2021	MGMT nonmethylated GBM	VEGF	anlotinib
Cell Cycle/Apoptosis/Transcription Pathways						
Study of the Poly (ADP-ribose) Polymerase-1 (PARP-1) Inhibitor BSI-201 in Patients with Newly Diagnosed Malignant Glioma	00687765	2008	2022	Newly diagnosed Malignant Glioma	PARP-1	iniparib (BSI-201) + TMZ + RT
Virus DNX2401 and Temozolomide in Recurrent Glioblastoma	01956734	2013	2017	GBM	Rb	DNX2401
Trial of Ponatinib in Patients with Bevacizumab-Refractory Glioblastoma	02478164	2013	2018	GBM	cKIT	Ponatinib
Safety and Efficacy of PD0332991 (Palbociclib), a Cyclin-dependent Kinase 4 and 6 Inhibitor, in Patients with Oligodendroglioma or Recurrent Oligoastrocytoma Anaplastic with the Activity of the Protein RB Preserved	02530320	2015	2020	Oligodendroma and oligoastrocytoma	CDK4/6	Palbociclib
Zotiraciclib (TG02) Plus Dose-Dense or Metronomic Temozolomide Followed by Randomized Phase II Trial of Zotiraciclib (TG02) Plus Temozolomide Versus Temozolomide Alone in Adults with Recurrent Anaplastic Astrocytoma and Glioblastoma	02942264	2016	2021	Glioma	CDK9	dinaciclib + TMZ
Phase I/IIa Study of Concomitant Radiotherapy with Olaparib and Temozolomide in Unresectable High Grade Gliomas Patients	03212742	2017	2023	Unresectable High Grade Glioma	poly(ADP-ribose) polymerase (PARP) inhibitor	olaparib + TMZ
A Phase 0/II Study of Ribociclib (LEE011) in Combination with Everolimus in Preoperative Recurrent High-Grade Glioma Patients Scheduled for Resection	03834740	2018	2023	Preoperative Recurrent High-Grade Glioma	CDK4/6, mTOR	ribociclib + everolimus
BGB-290 and Temozolomide in Treating Isocitrate Dehydrogenase (IDH)1/2-Mutant Grade I–IV Gliomas	03749187	2019	2023	Isocitrate Dehydrogenase (IDH)1/2-Mutant Grade I-IV Gliomas	Poly (ADP-Ribose) polymerase (PARP) inhibitor BGB-290	BGB-29 + TMZ
Anticancer Therapeutic Vaccination Using Telomerase-derived Universal Cancer Peptides in Glioblastoma	04280848	2020	2022	Primary GBM	TERT	UCPVax + anti-cancer vaccine based on the telomerase-derived helper peptides
B7-H3 CAR-T for Recurrent or Refractory Glioblastoma	04077866	2023	2022	Recurrent or refractory GBM	B7-H3	B7-H3 CAR-T
Immunotherapy Pathways						
A Dose Escalation and Cohort Expansion Study of Anti-CD27 (Varlilumab) and Anti-PD-1 (Nivolumab) in Advanced Refractory Solid Tumors	02335918	2015	2019	Refractory GBM	CD27, PD-1	varlilumab + nivolumab
Ipilimumab and/or Nivolumab in Combination with Temozolomide in Treating Patients with Newly Diagnosed Glioblastoma or Gliosarcoma	02311920	2015	2023	Newly diagnosed GBM	CTLA-4, PD-1	ipilimumab and/or nivolumab + TMZ
Study of Cabiralizumab in Combination with Nivolumab in Patients with Selected Advanced Cancers	02526017	2015	2022	Malignant Glioma	CSF1R TAMs, PD-1	cabiralizumab + nivolumab
Intra-tumoral Ipilimumab Plus Intravenous Nivolumab Following the Resection of Recurrent Glioblastoma	03233152	2016	2020	Recurrent GBM	CTLA-4, PD1	ipilimumab + nivolumab
Nivolumab for Recurrent or Progressive IDH Mutant Gliomas	03557359	2018	2022	Recurrent or Progressive IDH Mutant Gliomas	PD-1	Nivolumab
Efficacy and Safety of Pembrolizumab (MK-3475) Plus Lenvatinib (E7080/MK-7902) in Previously Treated Participants with Select Solid Tumors (MK-7902-005/E7080-G000-224/LEAP-005)	03797326	2019	2022	GBM	PD-1, multiple kinase inhibitors	Pembrolizumab, Lenvatinib
Efficacy of Nivolumab for Recurrent IDH Mutated High-Grade Gliomas	03925246	2019	2021	Recurrent IDH Mutated High-Grade Gliomas	PD-1	nivolumab
Trial of Anti-Tim-3 in Combination with Anti-PD-1 and SRS in Recurrent GBM	03961971	2020	2023	Recurrent GBM	TIM-3, PD-1	Sabatolimab, high-affinity, humanized, IgG4 (S228P) antibody + Spartalizumab + RT
Neoadjuvant Carilizumab and Apatinib for Recurrent High-Grade Glioma	04588987	2020	2020	Recurrent High-Grade Glioma	PD-1, TKI	carilizumab + apatinib
Ivosidenib (AG-120) with Nivolumab in IDH1 Mutant Tumors	04056910	2021	2023	IDH1 Mutant Tumors	IDH1, PD1	ivosidenib
Other						
A Phase 2b Clinical Study with a Combination Immunotherapy in Newly Diagnosed Patients with Glioblastoma	04485949	2023	2023	Newly diagnosed GBM	IGF1	IGV-001

Abbreviations: GBM, glioblastoma multiforme; EGFR, epithelial growth factor receptor; mTOR, mammalian target of rapamycin; CNS, central nervous system; PDGFR, platelet-derived growth factor receptor; TMZ, temozolomide; HER2, human epidermal growth factor receptor 2; RT, radiation therapy; IL2, interleukin-2; erbB, erythroblastic leukemia viral oncogene homologue; ALK, anaplastic lymphoma kinase; MAPK, mitogen-activated protein kinase; MEKi, mitogen-activated protein kinase kinase inhibitor; JAK/STAT, janus kinase-signal transducer and activator of transcription; PD-L1, programmed cell death ligand 1; GSK-3β, glycogen synthase kinase 3β; PI3K, phosphoinositide-3-kinase; tGLI1, truncated glioma-associated oncogene homolog-1; PARP-1, poly(ADP-ribose)-polymerase 1; Rb, retinoblastoma tumor suppressor; CDK, cyclin-dependent kinase; TERT, Telomerase reverse transcriptase; CTLA-4, cytotoxic T lymphocyte antigen 4; CSF1R TAMs, CSF1R-expressing tumor-associated macrophages; TIM-3, T-cell immunoglobulin and mucin domain 3; IGF1, insulin-like growth factor 1.

## Data Availability

Due to the nature of the research, there was no primary data collected. Materials were obtained from searches of the PubMed and Web of Science databases.

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
