# Peer review of "Systematic Review of Molecular Targeted Therapies for Adult-Type Diffuse Glioma: An Analysis of Clinical and Laboratory Studies"

_ijms, 2023, doi:10.3390/ijms241310456_

Round 1

Reviewer 1 Report

Review : Systematic Review of Molecular Targeted Therapies for Adult- Type Diffuse Glioma: An Analysis of Clinical and Laboratory  Studies

The review sought to describe actual literature on molecular targeted therapy in gliomas, considering the last 2021 WHO Classification of Tumors of the Central Nervous System.

The paper is interesting, clear, and with updated references, with the highlight of including both clinical and laboratory studies.

The only relevant limitation is that the wide variability in histologies and therapeutical strategies evaluated, makes really difficult to draw clear conclusions.

Nonetheless, the review gives a proper and comprehensive view of the current state of art on targeted therapy research in gliomas, thus providing a valid background for future clinical and preclinical trial.  

Author Response

Reviewer #1:
The review sought to describe actual literature on molecular targeted therapy in gliomas,
considering the last 2021 WHO Classification of Tumors of the Central Nervous System.
The paper is interesting, clear, and with updated references, with the highlight of including
both clinical and laboratory studies.
Response: We appreciate the reviewer’s positive review and are excited for the
enthusiasm towards our work.
Comment 1: The only relevant limitation is that the wide variability in histologies and
therapeutical strategies evaluated, makes really difficult to draw clear conclusions.

Response: We appreciate the reviewer for bringing this to our attention. To help clarify
this limitation in our study, we added some additional description of this limitation in the
discussion section. We hope this further elucidates that our study, while comprehensive
and broad in scope, is restricted by the vast methodological variation in both basic
science and clinical research on this topic.
Changes to Text: Our study, while comprehensive and broad in scope, is restricted by
the vast variation, particularly in histological methodology and molecular marker
identification capabilities.
(Page 12)
Comment 2: Nonetheless, the review gives a proper and comprehensive view of the current
state of art on targeted therapy research in gliomas, thus providing a valid background for
future clinical and preclinical trial.  
Response: Thank you for your comments. We hope these edits clarify and further
reinforce your support for this work.

Reviewer 2 Report

This paper aims at performing a systematic review of molecularly targeted therapies in diffuse gliomas. However, even after a rapid lecture, it emerges that several relevant published series/trials of molecularly targeted therapies in gliomas are missing (ex. PMID: 30522967, 28844499, 34088874, targeting of NTRK fusions, CheckMate 143 trial...), although published in the evaluated timeframe.

Additionally, as ongoing trials are not evaluated (no query of clinicaltrials.gov), there is a lack of the overview of ongoing studies.

Globally, the review has unacceptable omissions and does not add any relevant information to what is already published, and I would not recommend its publication.

Author Response

Reviewer #2:
This paper aims at performing a systematic review of molecularly targeted therapies in
diffuse gliomas.
Comment 1: However, even after a rapid lecture, it emerges that several relevant published
series/trials of molecularly targeted therapies in gliomas are missing
(ex. PMID: 30522967, 28844499, 34088874, targeting of NTRK fusions, CheckMate 143
trial...), although published in the evaluated timeframe.
Response: We appreciate you bringing this to our attention. Upon further review of the
systematic review methodology and search terms, it seems as though our initial search
terms were too narrow and failed to capture studies, such as the ones cited above, that do
not self-reference themselves as “molecular targeted therapy” studies. Given this
limitation, an extensive revision was conducted, beginning with the search terms. Instead
of solely searching “molecular targeted therapy” and “glioma,” the terms were broadened
to also include glioblastoma, ganglioglioma, and specific categories of molecular targeted
therapies that are prevalent in the literature on this topic, namely protein kinase
inhibitors, multikinase inhibitors, immune checkpoint inhibitors, antineoplastic combined
chemotherapy protocols.
With this adjustment, we were able to add 381 additional articles from PubMed and 105
additional studies in Web of Science. Particularly table 2, which denotes published
clinical trials on molecular targeted therapies in glioma, increased from 12 to 52. This
increase makes us feel more confident that we have more accurately captured published
clinical studies in this realm.
Changes to Text: The changes to the text were extensive. For brevity, the updated results
section, search terms, and PRISMA table are included below. Please refer to all
individual adjustments as highlighted in red for minor changes to qualitative data,
percentages, etc.
Results
This study identified 350 articles for inclusion. (Figure 1) Data extraction for each respective
category is detailed. Only 15% (52/350) of the total articles were clinical studies (Table 2). The
majority of articles (54%, 190/350) were laboratory studies investigating existing molecular
targeted therapies (Table 3), and 31% (108/350) were laboratory studies identifying new
molecular targets, without testing an existing therapy (Table 4). Across these groups, clinical
studies had a more recent median publication year (2017) compared to both laboratory studies
testing existing therapies (2015) and laboratory studies identifying novel targets (2016; p <
0.05).
(Page 5)
Search Terms
Appendix A. Search Terms

"Glioma" [Mesh] OR "Glioma/drug therapy" OR "Glioblastoma/drug therapy" OR
"Ganglioglioma/ drug therapy"[MeSH] OR “adult-type diffuse glioma” OR
“oligodendroglioma” OR “astrocytoma”
AND
“Molecular targeted Therapy” [MeSH] OR "Protein Kinase Inhibitors/administration
and dosage"[MeSH] OR "Antineoplastic Combined Chemotherapy
Protocols/administration and dosage" OR "Receptor Protein-Tyrosine Kinases/analysis"
OR "multikinase inhibitor" OR "MAP Kinase Kinase Kinases/antagonists and
inhibitors"[MeSH] OR "Mitogen-Activated Protein Kinase Kinases/antagonists and
inhibitors" OR "Immune Checkpoint Inhibitors/therapeutic use"**
**Format adapted for PubMed, Web of Science–Medline, and clinicaltrials.gov
advanced searches, respectively, from 01-01-1900 through 01-01-2023
(Page 64)
PRISMA Table

(Page 6)
Comment 2: Additionally, as ongoing trials are not evaluated (no query of
clinicaltrials.gov), there is a lack of the overview of ongoing studies.

Response: We appreciate and agree with the reviewer’s comment. We could have better
addressed the limitation. This suggestion prompted us to better explore what the current
state of research is in this area through a search clinicaltrials.gov with our same search
terms. This search yielded 341 studies, 119 of which met our inclusion criteria and were
what we defined as ongoing with a status of recruiting, active non recruiting, or
completed.
Changes to Text: The changes to the text are extensive. For brevity, the adjusted
methods, results, and discussion section are included below.
Methods
Additionally, a search with the same search terms was conducted on clinicaltrials.gov to assess
for clinical trials relating to molecular targeted therapy for adult-type diffuse glioma.
Data extraction for clinicaltrials.gov was limited to ongoing trials– defined as those with a status
of completed, recruiting or active, non-recruiting. The extracted variables included the following:
title of the study, the year started and year of most recent update, tumor type, NCT number, the
sponsoring or collaborating organization, the molecular target of interest, the intervention
utilized, as well as the phase of the study (Phase 1, 2, or 3), status (active or recruiting), funding
sources (NIH, industry, or other), and results, if available.
(Pages 3-4)
Results
A search of clincialtrials.gov yielded 341 clinical trials, of which 119 met our inclusion criteria
for ongoing clinical trials investigating molecular targeted therapies for adult-type diffuse glioma
and its subtypes. (Table 5) The most prevalent targets involved protein kinase pathways (65/119,
65%), followed by angiogenesis or microenvironmental targets (33/119, 28%), then cell
cycle/apoptosis (10/119, 8%) and immunotherapy pathways (10/119, 8%). For tumor types,
74/119 (62%) tested GBM, 5/119 (4%) tested IDH-mutant Astrocytoma, and 2/119 (2%) tested
oligodendroglioma, with many studies testing specific subcategories The average start year was
earlier in trials testing protein kinase targets (2009 +/- 6), compared with trials testing cell
cycle/apoptosis inhibitors (2016 +/- 4) and immunotherapies (2018 +/- 2), which occurred more
recently on average (p < 0.001).
The most common funding source was industry-related funding (54/119, 45%), followed by the
National Institute of Health (NIH) (45/119, 38%). (Figure S3) All ongoing clinical trials were
phase I or II.
(Page 8)
Discussion

Similarly to the published clinical studies on this topic, protein kinase pathways were by far the
most predominant molecular targets tested in ongoing clinical trials. Interestingly, these
therapeutics were also investigated much earlier on average. This finding is likely due to the fact
that protein kinase inhibitors are some of the earliest molecular target therapies in the field of
targeted oncologic interventions, thus able to start in clinical trials for the treatment of glioma as
early as 2001. 21 Perhaps, in the coming years, as the analysis of existing molecular targeted
therapies progresses from earlier stage clinical testing or laboratory testing, there will be a shift
favoring more of the scientifically novel approaches– such as immunotherapeutics, cell cycle
inhibitors, or more specifically localized targeting– in clinical trials.
(Page 10)
Please refer to Table 5 for the detailed data extraction and Supplementary Table for
more details regarding data extraction from clinicaltrials.gov.

Comment 2: Globally, the review has unacceptable omissions and does not add any
relevant information to what is already published, and I would not recommend its
publication.
Response: We appreciate the reviewer for bringing these limitations to our attention;
therefore, we have made major revisions to our search terms– prompting the screening of
486 additional articles on this topic, with detailed data extraction and review of 172
studies– and added 198 clinical trials with a search of clinicaltrials.gov to better capture
the question we that we initially intended to answer. We hope the reviewer agrees that
these revisions much better capture the state of molecular targeted therapy for glioma–
albeit still with limitations– but much more accurately.
Changes to Text: As described above– please refer to full-text changes, text colored in
red, for full clarification of adjustments

Reviewer 3 Report

Dear Authors,

Your systematic review on molecular targeted therapies is a very valuable contribution to the field of glioma research. Your very structured and comprehensive literature analysis ensures highest possible transparency of your used criteria. You also very clearly stated the pros and cons of this review. I also really appreciate the very strict adherence to the newest WHO classification which ensures that this review will be valid for a long time. Nevertheless, I do have some small suggestion how to improve further:

1.       Your analysis of experimental studies is impressive and dividing them into established and newly generated cell lines is very valuable. On point that is of interest here would be the percentage of studies that were performed on 3D cultures, as they are known to be even better than classical 2D cultures. Also if studies also already included organoids to get closer to the patient situation would be of interest, as these studies are thought to be more predictive.

2.       I don’t get the point of differentiating between “IDH-mutant glioma and oligodendroglioma”. To me oligodendroglioma are included in IDH-mutant glioma. Or do you want to differentiate IDH-mutant astrocytoma and oligodendroglioma. If so I would mention astrocytoma rather than IDH-mutant glioma.

3.       There are a few typos, double space or no space, in my version. I would recommend to carefully check for this before publishing.

Best regards

Author Response

Reviewer #3:
Dear Authors,
Your systematic review on molecular targeted therapies is a very valuable contribution to
the field of glioma research. Your very structured and comprehensive literature analysis
ensures highest possible transparency of your used criteria. You also very clearly stated the
pros and cons of this review. I also really appreciate the very strict adherence to the newest
WHO classification which ensures that this review will be valid for a long time.
Nevertheless, I do have some small suggestion how to improve further:
Response: We sincerely appreciate the reviewer’s overall support of our work.
Comment 1: Your analysis of experimental studies is impressive and dividing them into
established and newly generated cell lines is very valuable. On point that is of interest here
would be the percentage of studies that were performed on 3D cultures, as they are known
to be even better than classical 2D cultures. Also if studies also already included organoids
to get closer to the patient situation would be of interest, as these studies are thought to be
more predictive.
Response: We appreciate this suggestion and thus have implemented this analysis in our
study. Overall, we found
Changes to Text: (Page , Line )
Methods
To assess for 3-dimensional (3D) or spheroid technologies in laboratory studies of
existing therapies, a full text search of terms related to these technologies– “sphere,”
“spheroid,” “3D,” “3-D,” “3-Dimensional”– was also conducted.
(Page 4)
Results
Of the laboratory studies testing existing molecular targeted therapies, all were queried
for whether or not they utilized spheroid or 3-dimensional (3D) technologies for cell
culture as part of their methodology. Fifty-nine (32%) of studies adopted tumor sphere or
3D technology. (Table S2)
(Page 7)
Discussion
The use of technology such as spheroid or 3D cell culture is highly relevant in the context
of therapies for glioma. These technologies more accurately represent the tumor
microenvironment and allow for better design of patient-specific treatments. Nearly one-
third of laboratory studies testing existing therapies utilized this technology, implying
that these studies are likely closer to translation to human studies.
(Page 11)
Please refer to the added supplementary table 2 column for specific papers that fit this
new criterion.

Comment 2: I don’t get the point of differentiating between “IDH-mutant glioma and
oligodendroglioma”. To me oligodendroglioma are included in IDH-mutant glioma. Or do
you want to differentiate IDH-mutant astrocytoma and oligodendroglioma. If so I would
mention astrocytoma rather than IDH-mutant glioma.
Response: We appreciate the reviewer’s comment and agree that our initial draft added
confusion to the distinguishing of these IDH-mutant glioma types. We have edited it to
be clearer, with those that are IDH-mutant astrocytoma specifically labeled.
Changes to Text: All instances where “IDH-mutant Glioma” was utilized in the prior
manuscript was clarified to IDH-mutant astrocytoma, except when grouping the two
(astrocytoma and oligodendroglioma) specifically. Please refer to full text with changes
adjusted to red text to view these adjustments.
Comment 3: There are a few typos, double space or no space, in my version. I would
recommend to carefully check for this before publishing.
Response: Thank you for bringing this to our attention. Three authors carefully attended
to any typos or spacing incongruencies, reviewing the final draft more scrutinizingly.
Spelling errors, formatting inconsistencies in both the manuscript body and tables were
adjusted, and spacing made consistent between paragraphs.